# Learn the Time to Learn: Replay Scheduling for Continual Learning

## Abstract

Replay-based continual learning has shown to be successful in mitigating catastrophic forgetting. Most previous works focus on increasing the sample quality in the commonly small replay memory. However, in many real-world applications, replay memories would be limited by constraints on processing time rather than storage capacity as most organizations do store all historical data in the cloud. Inspired by human learning, we demonstrate that scheduling over which tasks to revisit is critical to the final performance with finite memory resources. To this end, we propose to learn the time to learn for a continual learning system, in which we learn schedules over which tasks to replay at different times using Monte Carlo tree search to illustrate this idea. We perform extensive evaluation and show that our method can learn replay schedules that significantly improve final performance across all tasks than baselines without considering the scheduling. Furthermore, our method can be combined with any other memory selection methods leading to consistently improved performance. Our results indicate that the learned schedules are also consistent with human learning insights.

## 1 Introduction

Many organizations deploying machine learning systems receive large volumes of data daily where these new data are often associated with new tasks. Although all historical data are stored in the cloud in practice, retraining machine learning systems on a daily basis is prohibitive both in time and cost. In this setting, the systems must continuously adapt to new tasks without forgetting the previously learned abilities. Continual learning methods (De Lange et al., 2019; McCloskey & Cohen, 1989; Parisi et al., 2019) address this challenge where, in particular, replay-based methods (Chaudhry et al., 2019; Hayes et al., 2020) have shown to be very effective in achieving great prediction performance and retaining knowledge of old tasks. Replay-based methods mitigate catastrophic forgetting by revisiting a small set of samples, which is feasible to process compared to the size of the historical data. In traditional continual learning literature, the replay memory is limited due to the assumption that historical data are not available. In the real-world setting where historical data are in fact always available, the requirement of small memory remains due to processing time and cost issues.

Most research on replay-based continual learning has been focused on the sample quality in the memory (Aljundi et al., 2019; Borsos et al., 2020; Chaudhry et al., 2019; Chrysakis & Moens, 2020; Nguyen et al., 2017; Rebuffi et al., 2017; Yoon et al., 2021) or data compression to increase the memory capacity (Hayes et al., 2020; Iscen et al., 2020; Pellegrini et al., 2019). Common for these methods is that the memory allocates an equal amount of space for storing samples from old tasks. When learning new tasks, the whole memory is replayed to mitigate catastrophic forgetting. However, in life-long learning settings, this simple strategy would be inefficient as the memory must store a large number of tasks. Furthermore, these methods ignore the time to learn old tasks again which is important in human learning. Humans are continual learning systems, and different methods have been developed to enhance memory retention, such as spaced repetition (Dempster, 1989; Ebbinghaus, 2013; Landauer & Bjork, 1977) which is used often in education. These education methods focus on the scheduling of learning and rehearsal of previous learned knowledge.

In this work, we argue that finding the proper schedule of what tasks to replay in the fixed memory setting is critical for continual learning. To demonstrate our claim, we perform a simple experiment

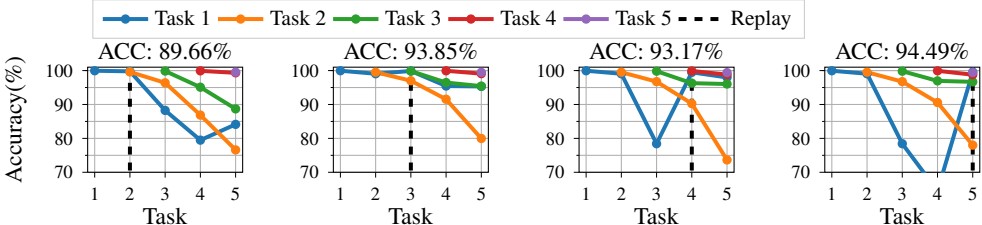

Figure 1: Task accuracies on Split MNIST (Zenke et al., 2017) when replaying only 10 samples of classes 0/1 at a single time step. The black vertical line indicates when replay is used. ACC denotes the average accuracy over all tasks after learning Task 5. Results are averaged over 5 seeds. These results show that the time to replay the previous task is critical for the final performance.

on the Split MNIST (Zenke et al., 2017) dataset where each task consists of learning the digits 0/1, 2/3, etc. arriving in sequence. The replay memory contains data from task 1 and can only be replayed at one point in time. Figure 1 shows how the task performances progress over time when the memory is replayed at different time steps. In this example, the best final performance is achieved when the memory is used when learning task 5. Note that choosing different time points to replay the same memory leads to noticeably different results in the final performance. These results indicate that scheduling the time when to apply replay can influence the final performance significantly of a continual learning system.

To this end, we propose learning the time to learn, in which we learn replay schedules of which tasks to replay at different times inspired from human learning (Dempster, 1989). We illustrate the advantages with replay scheduling by using Monte Carlo tree search (MCTS) (Coulom, 2006) to learn policies for replay. More specifically, we train a neural network on the current task dataset mixed with the scheduled replay samples and measure the final performance of the network to evaluate the replay schedules selected by MCTS. In summary, our contributions are:

- We demonstrate the importance of replay scheduling in continual learning and propose to learn the time to learn which tasks to replay (Section 3.1 and 3.2).

- We use MCTS as an example method to illustrate how replay schedules for continual learning can be learned by establishing a finite set of memory compositions that can be replayed at every task (Section 3.3).

- We demonstrate with several benchmark datasets that learned scheduling can improve the continual learning performance significantly in the fixed size memory setting (Section 4.1 and 4.4). Furthermore, we show that our method can be combined with any other memory selection methods (Section 4.3), as well as being efficient in situations where the memory size is even smaller than the number of classes (Section 4.5).

## 2 RELATED WORK

In this section, we give a brief overview of continual learning methods, essentially replay-based methods, as well as spaced repetition techniques for human continual learning.

**Continual Learning.** Traditional continual learning can be divided into three main areas, namely regularization-based, architecture-based, and replay-based approaches. Regularization-based methods aim to mitigate catastrophic forgetting by protecting parameters influencing the predictive performance from wide changes and use the rest of the parameters for learning the new tasks (Adel et al., 2019; Chaudhry et al., 2018a; Kirkpatrick et al., 2017; Li & Hoiem, 2017; Nguyen et al., 2017; Rannen et al., 2017; Schwarz et al., 2018; Zenke et al., 2017). Architecture-based methods isolate task-specific parameters by either increasing network capacity (Rusu et al., 2016; Yoon et al., 2019; 2017) or freezing parts of the network (Mallya & Lazebnik, 2018; Serra et al., 2018) to maintain good performance on previous tasks. Replay-based methods mix samples from old tasks with the current dataset to mitigate catastrophic forgetting, where the replay samples are either stored in an external memory (Chaudhry et al., 2019; Hayes et al., 2020; Isele & Cosgun, 2018; Lopez-Paz & Ranzato, 2017) or generated using a generative model (Shin et al., 2017; van de Ven & Tolias, 2018). Regularization-based approaches and dynamic architectures have been combined with replay-based approaches to methods to overcome their limitations (Chaudhry et al., 2018a;b; Douillard et al.,

2020; Ebrahimi et al., 2020; Joseph & Balasubramanian, 2020; Mirzadeh et al., 2020; Nguyen et al., 2017; Pan et al., 2020; Pellegrini et al., 2019; Rolnick et al., 2018; von Oswald et al., 2019). Our work relates most to replay-based methods with external memory which we spend more time on describing in the next paragraph.

**Replay-based Continual Learning.** A commonly used memory selection strategy of replay samples is random selection. Much research effort has focused on selecting higher quality samples to store in memory (Aljundi et al., 2019; Borsos et al., 2020; Chaudhry et al., 2019; Chrysakis & Moens, 2020; Hayes et al., 2019; Isele & Cosgun, 2018; Lopez-Paz & Ranzato, 2017; Nguyen et al., 2017; Rebuffi et al., 2017; Yoon et al., 2021). Chaudhry et al. (2019) reviews several selection strategies in scenarios with tiny memory capacity, e.g., reservoir sampling (Vitter, 1985), first-in first-out buffer (Lopez-Paz & Ranzato, 2017), k-Means, and Mean-of-Features (Rebuffi et al., 2017). However, more elaborate selection strategies have been shown to give little benefit over random selection for image classification problems (Chaudhry et al., 2018a; Hayes et al., 2020). More recently, there has been work on compressing raw images to feature representations to increase the number of memory examples for replay (Hayes et al., 2020; Iscen et al., 2020; Pellegrini et al., 2019). Our approach differs from the above mentioned works since we focus on learning to select which tasks to replay at the current task rather than improving memory selection or compression quality of the samples in the memory. Replay scheduling can however be combined with any selection strategy as well as storing feature representations.

**Human Continual Learning.** Humans are continual learning systems in the sense of learning tasks and concepts sequentially. Furthermore, humans have an impressive ability to memorize experiences but can forget learned knowledge gradually rather than catastrophically (French, 1999). Different learning techniques have been suggested for humans to memorize better (Dunlosky et al., 2013; Willis, 2007). An example is spaced repetition which gradually increases time-intervals between rehearsals for retaining long-term memory (Dempster, 1989). This technique has been studied frequently and was inspired from the works of Ebbinghaus (2013) on memory retention. For example, Landauer & Bjork (1977) demonstrated that memory training schedules using adjusted spaced repetition were better at preserving memory than uniformly spaced training. Hawley et al. (2008) studies the efficacy of spaced repetition on adults with probable Alzheimer's disease for learning face-name association. Several works in continual learning with neural networks are inspired by or have a connection to human learning techniques, including spaced repetition (Amiri, 2019; Amiri et al., 2017; Feng et al., 2019; Smolen et al., 2016), mechanisms of sleep (Ball et al., 2020; Mallya & Lazebnik, 2018; Schwarz et al., 2018), and reactivation of memories (Hayes et al., 2020; van de Ven et al., 2020). Our replay scheduling method is inspired by spaced repetition; we learn schedules of which memory samples to use for replay at different time steps.

## 3 METHOD

In this section, we describe our method for learning replay schedules for continual learning. The idea is to learn schedules of which memory examples the network should rehearse at different times. We use Monte Carlo tree search (MCTS) (Browne et al., 2012; Coulom, 2006) to learn a scheduling policy by encouraging searches for promising replay schedules based on the classification accuracy.

### 3.1 PROBLEM SETTING

We focus on the setting considering the real-world continual learning needs where all historical data are available but are prohibitively large. Therefore, only a small amount of historical data can be used when adapting the model to new data due to processing capability consideration. Thus, the goal is to learn how to select subsets of historical data to efficiently mitigate catastrophic forgetting when learning new tasks. We refer to these subsets of historical data as the *replay memory* throughout the paper, where the size of the replay memory affects the processing time when learning a new task $t$. Moreover, we focus on composing the replay memory based on the seen tasks in the historical data rather than single stored instances.

Next, we introduce the notation of our problem setting which resembles the traditional continual learning setting for image classification. We let a neural network $f_{\boldsymbol{\theta}}$, parameterized by $\boldsymbol{\theta}$, learn $T$ tasks sequentially given their corresponding task datasets $\mathcal{D}_1, \ldots, \mathcal{D}_T$ arriving in order. The $t$-th

dataset $\mathcal{D}_t = \{(\boldsymbol{x}_t^{(i)}, y_t^{(i)})\}_{i=1}^{N_t}$ consists of $N_t$ samples where $\boldsymbol{x}_t^{(i)}$ and $y_t^{(i)}$ are the $i$-th data point and class label respectively. The training objective at task $t$ is given by

$$\min_{\boldsymbol{\theta}} \sum_{i=1}^{N_t} \ell(f_{\boldsymbol{\theta}}(\boldsymbol{x}_t^{(i)}), y_t^{(i)}), \tag{1}$$

where $\ell(\cdot)$ is the loss function, e.g., cross-entropy loss in our case. Since $\mathcal{D}_t$ is only accessible at time step $t$, the network $f_{\boldsymbol{\theta}}$ is at risk of catastrophically forgetting the previous $t-1$ tasks when learning the current task. Replay-based continual learning methods mitigate the forgetting of old tasks by storing old examples in an external replay memory, that is mixed with the current task dataset during training. Next, we describe our method for constructing this replay memory.

We assume that historical data from old tasks are accessible at any time step $t$. However, since the historical data is prohibitively large, we can only fill a small replay memory $\mathcal{M}$ with $M$ historical samples for replay due to processing time constraints. The challenge is how to fill the replay memory with $M$ samples that efficiently retain the knowledge of old tasks when learning new tasks. We focus on selecting the samples on task-level by deciding on the task proportion $(a_1, \ldots, a_{t-1})$ of samples to fetch from each task, where $a_i \geq 0$ is the proportion of $M$ examples from task $i$ and $\sum_{i=1}^{t-1} a_i = 1$. Consequently, we need a method for choosing these task proportions of which old tasks to replay. To simplify this selection, we construct a discrete set of choices for possible task proportions telling how many samples from each task to use when constructing the replay memory $\mathcal{M}$.

## 3.2 Replay Scheduling for Continual Learning

In this section, we describe our replay scheduling method for selecting the replay memory at different time steps. We define a replay schedule as a sequence $S = (\boldsymbol{a}_1, \ldots, \boldsymbol{a}_{T-1})$, where $\boldsymbol{a}_i = (a_1, \ldots, a_{T-1})$ for $1 \leq i \leq T-1$, is the sequence of task proportions used for determining how many samples per task to fill the replay memory with at task $i$. To make the selection of task proportions tractable, we construct an action space with a discrete number of choices for the task proportions from old tasks. We use the following ways to construct this action space: At the time to learn task $t$, we have $t-1$ historical tasks that we can choose from. We create $t-1$ bins $\boldsymbol{b}_t = [b_1, b_2, \ldots b_{t-1}]$ and choose a task index to sample for each bin $b_i \in 1, \ldots, t-1$. We treat the bins as interchangeable and only keep the unique choices. For example, at task 3, we have seen task 1 and 2; so the unique choices of vectors are $[1, 1], [1, 2], [2, 2]$, where $[1, 1]$ indicates that all memory samples are from task 1, $[1, 2]$ indicates that half memory is from task 1 and the other half are from task etc. The task proportion are then computed by counting the number of occurrences of each task index in $\boldsymbol{b}_t$ and dividing by the $t-1$, such that $\boldsymbol{a}_t = \texttt{bincount}(\boldsymbol{b}_t)/(t-1)$. If the memory size $M$ is not divisible by the task proportion value for task $i$, we round the number of replay samples from task $i$, i.e., $a_i \cdot M$, up or down accordingly while keeping the memory size fixed. From this specification, we can build a tree of different replay schedules to evaluate with the network.

Figure 2 shows an example of such a replay schedule tree with Split MNIST (Zenke et al., 2017) where the memory size has been set to $M = 8$. The figure shows the current tasks with an example image of the task classes on the left, and the right side shows examples of possible replay memories that can be evaluated. The memory starts as the empty set, i.e. $\mathcal{M}_1 = \emptyset$, at task 1. Before learning task 2, $\mathcal{M}_2$ is filled with $M$ task 1 examples since this is the only task seen so far. At task 3, the memory compositions we can choose from are $M$ examples from either task 1 or 2, as well as equally filling $\mathcal{M}_3$ with four examples each from both tasks. A replay schedule is represented as a path traversal of different replay memory compositions from task 1 to task 5. We have color-coded three examples of possible schedules in Figure 2 to use for illustration: the blue path represents a replay schedule where only task 1 examples are replayed at all future tasks. The red path represents using an equally distributed amount of memory samples per task in the memory, and the purple path represents a schedule where the memory is filled $M$ examples from the previously visited task. Note that all other possible paths in the tree are also valid replay schedules.

## 3.3 Monte Carlo Tree Search for Replay Schedules

Our proposed setup to find replay schedules is to discretize the number of possible task proportions per task. The action space of task proportions then turn into a tree shape where each node represents

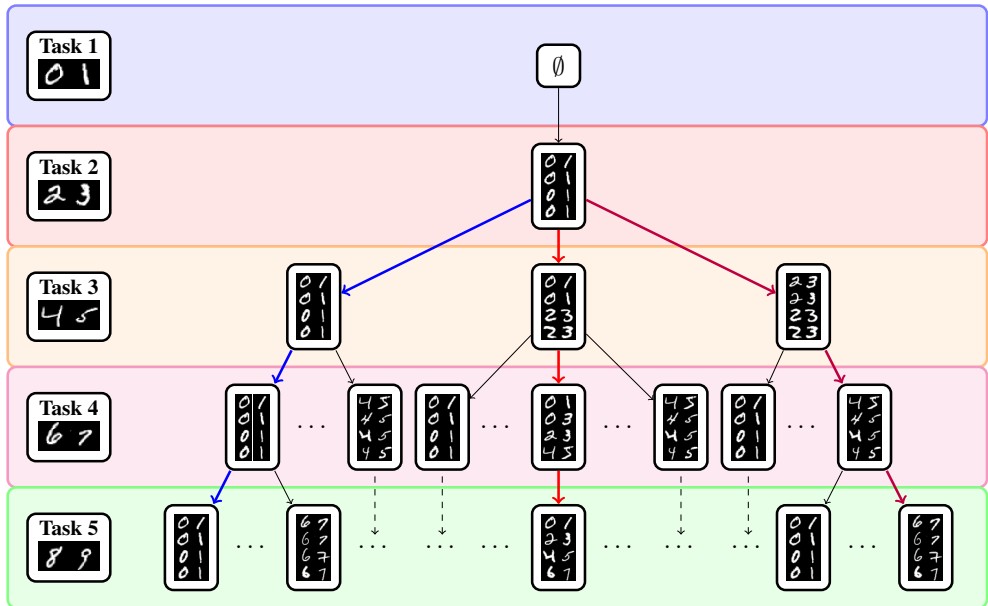

Figure 2: An exemplar tree of replay memory compositions from the proposed discretization method described in Section 3.2 for Split MNIST. The replay memories from one replay schedule are found by traversing from task 1-5 through the tree on the right hand side. The replay memory compositions have been structured according to the task where they can be used for replay. Note that the replay memory at task 1 is the empty set, i.e., $\mathcal{M} = \emptyset$. Example images for each task are shown on the left.

a task proportion that corresponds to a specific composition of the replay memory $\mathcal{M}$, as can be seen in Figure 2. To find the efficient replay schedule $S$ in mitigating catastrophic forgetting, we can perform an exhaustive search (e.g., breadth-first search) where all possible replay schedules are evaluated. However, since the tree grows fast with the number of tasks, we need a scalable method that enables tree searches in large search spaces. To this end, we propose to use Monte Carlo tree search (MCTS) (Browne et al., 2012; Coulom, 2006) since MCTS has been successful in applications with similar conditions (Chaudhry & Lee, 2018; Gelly et al., 2006; Silver et al., 2016). In our setting, MCTS can help us to concentrate the search in directions towards promising replay schedules in terms of classification performance and thus learn the time to learn.

Before outlining the steps for performing MCTS, we describe the notation for the tree search. The tree has $T$ levels where level $t$ corresponds to task $t$. Each tree level $t$ contains a set of nodes $V_t = \{v_t^i\}_{i=1}^{K_t}$ where $K_t$ is the number of nodes at level $t$. Every node $v_t^i$ has a corresponding sequence of task proportions $\boldsymbol{a}_t^i$, which can be used to retrieve the replay memory from the historical data. Referring back to Figure 2, the replay memory composition on the root node $v_1^1$ is the empty set $\mathcal{M} = \emptyset$, since the historical data is empty. At task 2, the only task proportion is $\boldsymbol{a}_2^1 = (a_1, a_2, ..., a_{T-1}) = (1.0, 0.0, ..., 0.0)$, so the replay memory at node $v_2^1$ is only filled with samples from task 1. In the rest of the paper, we denote a node at task $t$ as $v_t$ by ignoring the superscript $i$ to avoid clutter in the notation.

At every visited node $v_t$, we store the corresponding task proportions $\boldsymbol{a}_t$ in the replay schedule $S$. The final replay schedule is then used for constructing the replay memory at each time step $t$ during the continual learning training. Next, we briefly outline the steps for performing MCTS in the search for replay schedules.

**Selection.** An MCTS iteration begins by selecting the next node to visit from the root node. If the current node $v_t$ has visited all its children during the previous iterations, the next node is selected by evaluating the Upper Confidence Tree (UCT) (Kocsis & Szepesvári, 2006) function for all children. We use the following UCT function proposed by Chaudhry & Lee (2018) to evaluate the score for moving from node $v_t$ to its child $v_{t+1}$:

$$UCT(v_t, v_{t+1}) = \max(Q(v_{t+1})) + C\sqrt{\frac{2\log(N(v_t))}{N(v_{t+1})}}. \tag{2}$$

The reward function $Q(\cdot)$ contains the rewards for previous searches that has passed through the child $v_{t+1}$. The exploration constant $C \geq 0$ determines the degree of exploration of less visited replay schedules based on the number of visits $N(v_t)$ and $N(v_{t+1})$ to the corresponding nodes in the tree. The child node to visit next is the one with the highest UCT score.

**Expansion.** Whenever the current node $v_t$ has unvisited child nodes, the search tree is expanded with one of the unvisited child nodes $v_{t+1}$ selected with uniform sampling.

**Simulation and Reward.** After the expansion step, the search proceeds by selecting the succeeding nodes uniformly at random until a terminal node $v_T$ is reached. The replay schedule $S$ is then collected by appending the task proportions from the visited nodes during the iteration. We train the network with the replay memories retrieved from $S$ and evaluate the average accuracy over all tasks after learning the final task $T$ as the reward $r$, i.e., $r = \frac{1}{T} \sum_{i=1}^{T} A_{T,i}^{(val)}$, where $A_{T,i}^{(val)}$ is the validation accuracy of task $i$ after learning task $T$.

**Backpropagation.** The reward $r$ is backpropagated from the expanded node $v_t$ up to the root node $v_1$, where the number of visits $N(\cdot)$ and reward function $Q(\cdot)$ for each node is updated.

We provide pseudocode in Algorithm 1 in Appendix B. Finally, we emphasize that we use MCTS to illustrate in a simple setting the importance of learning the time to learn old tasks again.

## 4 EXPERIMENTS

We evaluated our replay scheduling method empirically on six common benchmark datasets for continual learning. We show that scheduling of the replay memory improves significantly over replaying equal proportions across the tasks. Furthermore, we demonstrate that replay scheduling can be as efficient as replaying all available memory samples in settings where only 1 example per class can be stored for replay. We denote our method as Replay Scheduling MCTS (RS-MCTS) and select the result on the held-out test sets from the replay schedule that yielded the best reward on the validation set during the search. We perform experiments using 5 different seeds on all datasets. Our code is available as part of the supplementary material. The full details on experimental settings are in Appendix A and additional results are in Appendix D.

**Datasets.** We conduct experiments on six datasets commonly used as benchmarks in the continual learning literature: Split MNIST (LeCun et al., 1998; Zenke et al., 2017), Fashion-MNIST (Xiao et al., 2017), Split notMNIST (Bulatov, 2011), Permuted MNIST (Goodfellow et al., 2013), Split CIFAR-100 (Krizhevsky & Hinton, 2009), and Split miniImagenet (Vinyals et al., 2016). We randomly sample 15% of the training data from each task to use for validation when computing the reward for the MCTS simulations.

**Baselines.** We compare RS-MCTS to using 1) random replay schedules (Random), 2) equal task schedules (ETS), and 3) a heuristic scheduling method (Heuristic). The ETS baseline uses equal task proportions, such that $M/(t - 1)$ samples per task are replayed during learning of task $t$, and use both training and validations sets for training such that they use the same amount of data as RS-MCTS. The Heuristic baseline replays the old tasks which accuracy on the validation set is below a certain threshold proportional to the best achieved validation accuracy on the task. Here, the replay memory is filled with $M/k$ where $k$ is the number of tasks that need to be replayed according to their decrease in validation accuracy. See Appendix C for more details on the Heuristic baseline.

**Network Architectures.** We use a multi-head output layer for all datasets except for Permuted MNIST where the network uses single-head output layer. We use a 2-layer MLP with 256 hidden units for Split MNIST, Split FashionMNIST, Split notMNIST, and Permuted MNIST. For Split CIFAR-100, we use the CNN architecture used in Vinyals et al. (2016) and Schwarz et al. (2018). For Split mniImagenet, we apply the reduced ResNet-18 from Lopez-Paz & Ranzato (2017).

**Evaluation Metric.** We use the average test accuracy over all tasks after learning the final task, i.e., ACC $= \frac{1}{T} \sum_{i=1}^{T} A_{T,i}$ where $A_{T,i}$ is the test accuracy of task $i$ after learning task $T$.

### 4.1 RESULTS ON REWARD FOR RS-MCTS

In the first experiments, we show that the replay schedules from RS-MCTS yield better performance than replaying an equal amount of samples per task. The replay memory size is fixed to $M = 10$ for Split MNIST, FashionMNIST, and notMNIST, and $M = 100$ for Permuted MNIST, Split CIFAR-100, and Split miniImagenet. Uniform sampling is used as the memory selection method for all

NEW
(dhXe)

NEW
(8z4B)

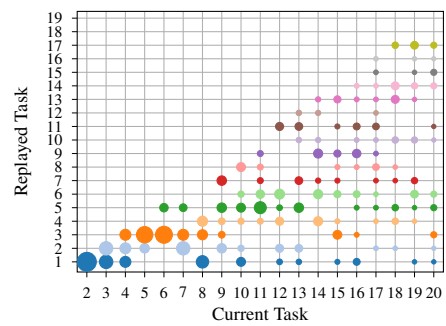

Figure 3: Average test accuracies over tasks after learning the final task (ACC) over the MCTS simulations for all datasets, where 'S' and 'P' are used as short for 'Split' and 'Permuted'. We compare performance for RS-MCTS (Ours) against random replay schedules (Random), Equal Task Schedule (ETS), and Heuristic Scheduling (Heuristic) baselines. For the first three datasets, we show the best ACC found from a breadth-first search (BFS) as an upper bound. All results have been averaged over 5 seeds. These results show that replay scheduling can improve significantly over ETS and outperform or perform on par with Heuristic across different datasets and network architectures.

methods in this experiment. For the 5-task datasets, we provide the optimal replay schedule found from a breadth-first search (BFS) over all possible replay schedules in our action space as an upper bound for RS-MCTS. As the search space grows fast with the number of tasks, it will even with only 5 continual learning tasks (which corresponds to a tree with depth of 4) yield a tree with 1050 leaf nodes. Thus, BFS becomes computationally infeasible when we have 10 or more tasks.

Figure 3 shows the progress of ACC over the MCTS iterations by RS-MCTS for all datasets. We also show the best ACC metrics for ETS, Heuristic, and BFS as straight lines. We observe that RS-MCTS outperforms Random and ETS successively with more iterations. Furthermore, RS-MCTS approaches the upper limit from BFS on the 5-task datasets. For Permuted MNIST and Split CIFAR-100, the Heuristic baseline and RS-MCTS perform on par after 50 iterations. This shows that Heuristic with careful tuning of the validation accuracy threshold can be a strong baseline when comparing against replay scheduling. The top row of Table 1 shows the ACC for each method for this experiment. We note that RS-MCTS outperforms ETS significantly for most datasets and performs on par with Heuristic.

NEW
(8z4B)

## 4.2 REPLAY SCHEDULE VISUALIZATIONS

We visualize a learned replay schedule from Split CIFAR-100 with memory size $M = 100$ to gain insights into the behavior of the scheduling policy from RS-MCTS. Figure 4 shows a bubble plot of the task proportions that are used for filling the replay memory at every task. Each color of the circles corresponds to a historical task and its size represents the proportion of examples that are replayed at the current task. The sum of all points from all circles at each column is fixed at the different time steps since the memory size $M$ is fixed. The task proportions vary dynamically over time in a sophisticated nonlinear way which would be hard to replace by a heuristic method. Moreover, we can observe space repetition style scheduling in many tasks, e.g., task 1-3 are replayed with similar pro-

Figure 4: Visualization using a bubble plot of a replay schedule learned from Split CIFAR-100. The task proportions vary dynamically over time which would be hard to replace by a heuristic method.

portion at the initial tasks but eventually starts varying the time interval between replay. Also, task 4 and 6 need less replay in their early stages, which could potentially be that they are simpler or correlated with other tasks. We provide a similar visualization for Split MNIST in Figure 7 in Appendix D.1 to bring more insights to the benefits of replay scheduling.

NEW
(kfup)

## 4.3 ALTERNATIVE MEMORY SELECTION METHODS

In this section, we show that our method can be combined with any memory selection method for storing replay samples. In addition to uniform sampling, we apply various memory selection methods commonly used in the continual learning literature, namely $k$-means clustering, $k$-center clustering (Nguyen et al., 2017), and Mean-of-Features (MoF) (Rebuffi et al., 2017). We compare

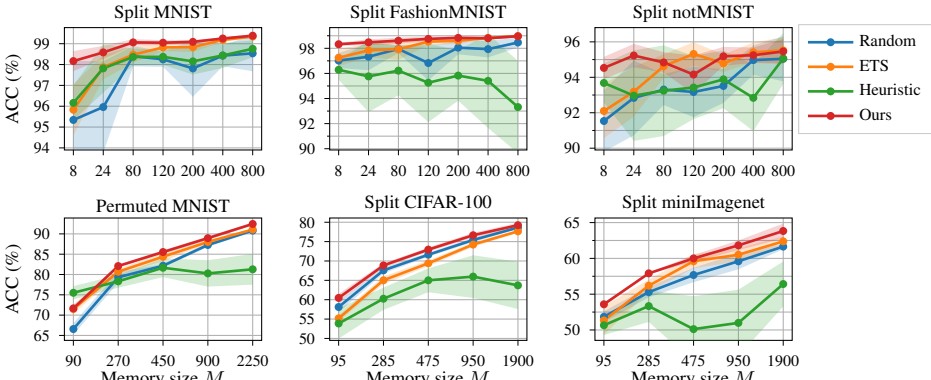

Figure 5: Average test accuracies over tasks after learning the final task (ACC) over different replay memory sizes $M$ for the RS-MCTS (Ours) and the Random, ETS, and Heuristic baselines on all datasets. All results have been averaged over 5 seeds. The results show that replay scheduling can outperform replaying with random and equal task proportions, especially for small $M$, on both small and large datasets across different backbone choices. Furthermore, our method requires less careful tuning than the Heuristic baseline as $M$ increases.

Table 1: Accuracy comparison between RS-MCTS (Ours), Random choice scheduling (Random), Heuristic Scheduling (Heuristic), and Equal Task Schedule (ETS) with various memory selection methods evaluated across all datasets. We use 'S' and 'P' as short for 'Split' and 'Permuted' for the datasets. The replay memory size is $M = 10$ and $M = 100$ for the 5-task and 10/20-task datasets respectively. We report the mean and standard deviation averaged over 5 seeds. RS-MCTS performs better or on par than the baselines on most datasets and selection methods, where MoF yields the best performance in general.

| Selection | Method | 5-task Datasets | | | 10- and 20-task Datasets | | |
| | | S-MNIST | S-FashionMNIST | S-notMNIST | P-MNIST | S-CIFAR-100 | S-miniImagenet |
|---|---|---|---|---|---|---|---|
| Uniform | Random | $95.91 \pm 1.56$ | $95.82 \pm 1.45$ | $92.39 \pm 1.29$ | $72.44 \pm 1.15$ | $53.99 \pm 0.51$ | $48.08 \pm 1.36$ |
| | ETS | $94.02 \pm 4.25$ | $95.81 \pm 3.53$ | $91.01 \pm 1.39$ | $71.09 \pm 2.31$ | $47.70 \pm 2.16$ | $46.97 \pm 1.24$ |
| | Heuristic | $96.02 \pm 2.32$ | $97.09 \pm 0.62$ | $91.26 \pm 3.99$ | $76.68 \pm 2.13$ | $57.31 \pm 1.21$ | $49.66 \pm 1.10$ |
| | Ours | $97.93 \pm 0.56$ | $98.27 \pm 0.17$ | $94.64 \pm 0.39$ | $76.34 \pm 0.98$ | $56.60 \pm 1.13$ | $50.20 \pm 0.72$ |
| $k$-means | Random | $94.24 \pm 3.20$ | $96.30 \pm 1.62$ | $91.64 \pm 1.39$ | $74.30 \pm 1.43$ | $53.18 \pm 1.66$ | $49.47 \pm 2.70$ |
| | ETS | $92.89 \pm 3.53$ | $96.47 \pm 0.85$ | $93.80 \pm 0.82$ | $69.40 \pm 1.32$ | $47.51 \pm 1.14$ | $45.82 \pm 0.92$ |
| | Heuristic | $96.28 \pm 1.68$ | $95.78 \pm 1.50$ | $91.75 \pm 0.94$ | $75.57 \pm 1.18$ | $54.31 \pm 3.94$ | $49.25 \pm 1.00$ |
| | Ours | $98.20 \pm 0.16$ | $98.48 \pm 0.26$ | $93.61 \pm 0.71$ | $77.74 \pm 0.80$ | $56.95 \pm 0.92$ | $50.47 \pm 0.85$ |
| $k$-center | Random | $96.40 \pm 0.68$ | $95.57 \pm 3.16$ | $92.61 \pm 1.70$ | $71.41 \pm 2.75$ | $48.46 \pm 0.31$ | $44.76 \pm 0.96$ |
| | ETS | $94.84 \pm 1.40$ | $97.28 \pm 0.50$ | $91.08 \pm 2.48$ | $69.11 \pm 1.69$ | $44.13 \pm 1.06$ | $41.35 \pm 1.23$ |
| | Heuristic | $94.55 \pm 2.79$ | $94.08 \pm 3.72$ | $92.06 \pm 1.20$ | $74.33 \pm 2.00$ | $50.32 \pm 1.97$ | $44.13 \pm 0.95$ |
| | Ours | $98.24 \pm 0.36$ | $98.06 \pm 0.35$ | $94.26 \pm 0.37$ | $76.55 \pm 1.16$ | $51.37 \pm 1.63$ | $46.76 \pm 0.96$ |
| MoF | Random | $95.18 \pm 3.18$ | $95.76 \pm 1.41$ | $91.33 \pm 1.75$ | $77.96 \pm 1.84$ | $61.93 \pm 1.05$ | $54.50 \pm 1.33$ |
| | ETS | $97.04 \pm 1.23$ | $96.48 \pm 1.33$ | $92.64 \pm 0.87$ | $77.62 \pm 1.12$ | $60.43 \pm 1.17$ | $56.12 \pm 1.12$ |
| | Heuristic | $96.46 \pm 2.41$ | $95.84 \pm 0.89$ | $93.24 \pm 0.77$ | $77.27 \pm 1.45$ | $55.60 \pm 2.70$ | $52.30 \pm 0.59$ |
| | Ours | $98.37 \pm 0.24$ | $97.84 \pm 0.32$ | $94.62 \pm 0.42$ | $81.58 \pm 0.75$ | $64.22 \pm 0.65$ | $57.70 \pm 0.51$ |

our method and ETS combined with these different selection methods. As in Section 4.1, we set the replay memory size $M = 10$ for the 5-task datasets and $M = 100$ for the 10- and 20-task datasets. Table 1 shows the results across all datasets. We note that using the replay schedule from RS-MCTS outperforms the baselines when using the alternative selection methods, where MoF performs the best on most datasets.

NEW
(8z4B)

## 4.4 RESULTS WITH VARYING REPLAY MEMORY SIZE

We show that our method can improve the performance across different choices of memory size. In this experiment, we set the replay memory size to ensure the ETS baseline replays an equal number of samples per class at the final task. The memory size is set to $M = n_{cpt} \cdot n_{spc} \cdot (T - 1)$, where $n_{cpt}$ is the number of classes per task in the dataset and $n_{spc}$ are the number of samples per class we wish to replay at task $T$ for the ETS baseline. In Figure 5, we observe that RS-MCTS yield better task accuracies than ETS, especially for small memory sizes. Both RS-MCTS and ETS perform better than Heuristic as $M$ increases showing that Heuristic requires careful tuning of the validation accuracy threshold.

FIX
(8z4B)

Table 2: Accuracy comparison in the 1 example per class memory setting evaluated across all datasets. We use 'S' and 'P' as short for 'Split' and 'Permuted' for the datasets. RS-MCTS has replay memory size $M = 2$ and $M = 50$ for the 5-task and 10/20-task datasets respectively. The baselines replay all available memory samples. We report the mean and standard deviation averaged over 5 seeds. RS-MCTS performs on par with the best baselines on all datasets except S-CIFAR-100.

| Method | 5-task Datasets | | | 10- and 20-task Datasets | | |
| --- | --- | --- | --- | --- | --- | --- |
| | S-MNIST | S-FashionMNIST | S-notMNIST | P-MNIST | S-CIFAR-100 | S-miniImagenet |
| Random | $92.56 \pm 2.90$ | $92.70 \pm 3.78$ | $89.53 \pm 3.96$ | $70.02 \pm 1.76$ | $48.62 \pm 1.02$ | $48.85 \pm 1.38$ |
| A-GEM | $94.97 \pm 1.50$ | $94.81 \pm 0.86$ | $92.27 \pm 1.16$ | $64.71 \pm 1.78$ | $42.22 \pm 2.13$ | $32.06 \pm 1.83$ |
| ER-Ring | $94.94 \pm 1.56$ | $95.83 \pm 2.15$ | $91.10 \pm 1.89$ | $69.73 \pm 1.13$ | $53.93 \pm 1.13$ | $49.82 \pm 1.69$ |
| Uniform | $95.77 \pm 1.12$ | $97.12 \pm 1.57$ | $92.14 \pm 1.45$ | $69.85 \pm 1.01$ | $52.63 \pm 1.62$ | $50.56 \pm 1.07$ |
| RS-MCTS (Ours) | $96.07 \pm 1.60$ | $97.17 \pm 0.78$ | $93.41 \pm 1.11$ | $72.52 \pm 0.54$ | $51.50 \pm 1.19$ | $50.70 \pm 0.54$ |

## 4.5 EFFICIENCY OF REPLAY SCHEDULING

Here, we illustrate the efficiency of replay scheduling with comparisons to several common replay-based continual learning baselines in an even more extreme memory setting. Our goal is to investigate if scheduling over which tasks to replay can be more efficient in situations where the memory size is even smaller than the number of classes. To this end, we set the replay memory size for our method to $M = 2$ for the 5-task datasets, such that only 2 samples can be selected for replay at all times. For the 10- and 20-task datasets which have 100 classes, we set $M = 50$. We then compare against the most memory efficient continual learning baselines, namely A-GEM (Chaudhry et al., 2018b), ER-Ring (Chaudhry et al., 2019) which show promising results with 1 sample per class for replay after learning each task, and with uniform memory selection as reference. Additionally, we compare to using random replay schedules (Random) with the same memory setting as for RS-MCTS. We visualize the memory usage when training on one of the 5-task datasets for our method and the baselines in Figure 6. We also visualize the memory usages for the other benchmarks in the Appendix E.

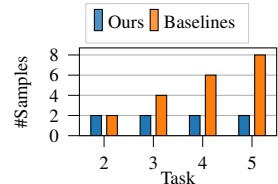

Figure 6: Number of replayed samples per task for the 5-task datasets in our tiny memory setting. Our method uses a fixed $M = 2$ samples for replay, while the baselines increment their memory per task.

Table 2 shows the ACC for each method across all datasets. Despite using significantly fewer samples for replay, RS-MCTS performs better or on par with the best baselines on all datasets except Split CIFAR-100. These results show that replay scheduling can be even more efficient than the state-of-the-art memory efficient replay-based methods, which indicate that learning the time to learn is an important research direction in continual learning.

## 5 CONCLUSIONS

We propose learning the time to learn, i.e., in a real-world continual learning context, learning schedules of what previous tasks to replay at different times. To the best of our knowledge, we are the first to consider the time to learn in machine learning inspired by human learning techniques. We demonstrate with an example method that learned replay schedules produce significantly improved results under the same memory budget when comparing with the method without scheduling. Furthermore, the dynamic behavior of the learned schedules showed similarities to human learning techniques, such as spaced repetition, by rehearsing previous tasks with varying time intervals. Finally, we showed that replay scheduling allows for utilizing the memory more efficiently compared to standard benchmarks replaying all memory samples in the tiny memory setting.

In future work, we would like to explore choosing memory samples on an instance level as the current work selects samples on task level. This would need a policy learning method, that scales to large action spaces which is a research challenge by itself. Also, our demonstrated method with MCTS can be inefficient, especially in continual learning settings, in terms of training time and since it needs to learn for each dataset and application separately. In future work, we will incorporate reinforcement learning methods that generalize. We will therefore extend replay scheduling to learn general policies that can be directly applied to any new application and domain.

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

APPENDIX

This supplementary material is structured as follows:

- Appendix A: Full details of the experimental settings.
- Appendix B: Pseudocode of RS-MCTS in Algorithm 1 to provide more details about our method.
- Appendix C describes the Heuristic scheduling baseline.
- Appendix D shows additional experimental results.
- Appendix E includes additional figures.

## A   EXPERIMENTAL SETTINGS

In this section, we describe the full details of the experimental settings used in this paper.

**Datasets:**   We conduct experiments on six datasets commonly used in the continual learning literature. Split MNIST (Zenke et al., 2017) is a variant of the MNIST (LeCun et al., 1998) dataset where the classes have been divided into 5 tasks incoming in the order 0/1, 2/3, 4/5, 6/7, and 8/9. Split Fashion-MNIST (Xiao et al., 2017) is of similar size to MNIST and consists of grayscale images of different clothes, where the classes have been divided into the 5 tasks T-shirt/Trouser, Pullover/Dress, Coat/Sandals, Shirt/Sneaker, and Bag/Ankle boots. Similar to MNIST, Split notM-NIST (Bulatov, 2011) consists of 10 classes of the letters A-J with various fonts, where the classes are divided into the 5 tasks A/B, C/D, E/F, G/H, and I/J. We use training/test split provided by Ebrahimi et al. (2020) for Split notMNIST. Permuted MNIST Goodfellow et al. (2013) dataset consists of applying a unique random permutation of the pixels of the images in original MNIST to create each task, except for the first task that is to learn the original MNIST dataset. We reduce the original MNIST dataset to 10k samples and create 9 unique random permutations to get a 10-task version of Permuted MNIST. In Split CIFAR-100 (Krizhevsky & Hinton, 2009), the 100 classes are divided into 20 tasks with 5 classes for each task (Lopez-Paz & Ranzato, 2017; Rebuffi et al., 2017). Similarly, Split miniImagenet Vinyals et al. (2016) consists of 100 classes randomly chosen from the original Imagenet dataset where the 100 classes are divided into 20 tasks with 5 classes per task.

**Network Architectures:**   We use a 2-layer MLP with 256 hidden units and ReLU activation for Split MNIST, Split FashionMNIST, Split notMNIST, and Permuted MNIST. We use a multi-head output layer for each dataset except Permuted MNIST where the network uses single-head output layer. For Split CIFAR-100, we use a multi-head CNN architecture built according to the CNN in (Adel et al., 2019; Schwarz et al., 2018; Vinyals et al., 2016), which consists of four 3x3 convolutional blocks, i.e. convolutional layer followed by batch normalization (Ioffe & Szegedy, 2015), with 64 filters, ReLU activations, and 2x2 Max-pooling. For Split mniImagenet, we use the reduced ResNet-18 from Lopez-Paz & Ranzato (2017) with multi-head output layer.

**Hyperparameters:**   We train all networks with the Adam optimizer (Kingma & Ba, 2014) with starting learning rate $\eta = 0.001$ and hyperparameters $\beta_1 = 0.9$ and $\beta_2 = 0.999$. Note that the learning rate for Adam is not reset before training on a new task. Next, we give details on number of training epochs and batch sizes specific for each dataset:

- Split MNIST: 10 epochs/task, batch size 128.
- Split FashionMNIST: 30 epochs/task, batch size 128.
- Split notMNIST: 50 epochs/task, batch size 128.
- Permuted MNIST: 20 epochs/task, batch size 128.
- Split CIFAR-100: 25 epochs/task, batch size 256.
- Split miniImagenet: 1 epoch/task (task 1 trained for 5 epochs as warm up), batch size 32.

**Monte Carlo Tree Search:**   We run RS-MCTS for 100 iterations in all experiments. The replay schedules used in the reported results on the held-out test sets are from the replay schedule that gave the highest reward on the validation sets. The exploration constant for UCT in Equation 2 is set to $C = 0.1$ in all experiments (Chaudhry & Lee, 2018).

**Computational Cost:**  All experiments were performed on one NVIDIA GeForce RTW 2080Ti. The wall clock time for ETS on Split MNIST was around 1.5 minutes, and RS-MCTS and BFS takes 40 seconds on average to run one iteration, where BFS runs 1050 iterations in total for Split MNIST.

**Implementations:**  We adapted the implementation released by Borsos et al. (2020) for the memory selection strategies Uniform sampling, $k$-means clustering, $k$-center clustering (Nguyen et al., 2017), and Mean-of-Features (Rebuffi et al., 2017). Furthermore, we follow the implementations released by Chaudhry et al. (2019) and Mirzadeh & Ghasemzadeh (2021) for A-GEM (Chaudhry et al., 2018b) and ER-Ring (Chaudhry et al., 2019).

**Experimental Settings for with Single Task Replay Memory Experiment:**  We motivated the need for replay scheduling in continual learning with Figure 1 in Section 1. This simple experiment was performed on Split MNIST where the replay memory only contains samples from the first task, i.e., learning the classes 0/1. Furthermore, the memory can only be replayed at one point in time and we show the performance on each task when the memory is replayed at different time steps. We set the memory size to $M = 10$ samples such that the memory holds 5 samples from both classes. We use the same network architecture and hyperparameters as described above for Split MNIST. The ACC metric above each subfigure corresponds to the ACC for training a network with the single task memory replay at different tasks. We observe that choosing different time points to replay the same memory leads to noticeably different results in the final performance, and in this example, the best final performance is achieved when the memory is used when learning task 5. Therefore, we argue that finding the proper schedule of what tasks to replay at what time in the fixed memory situation can be critical for continual learning.

## B  REPLAY SCHEDULING MONTE CARLO TREE SEARCH ALGORITHM

We provide pseudo-code in Algorithm 1 outlining the steps for our method Replay Scheduling Monte Carlo tree search (RS-MCTS) described in the main paper (Section 3.3). The MCTS procedure selects actions over which task proportions to fill the replay memory with at every task, where the selected task proportions are stored in the replay schedule $S$. The schedule is then passed to EVALUATEREPLAYSCHEDULE($\cdot$) where the continual learning part executes the training with replay memories filled according to the schedule. The reward for the schedule $S$ is the average validation accuracy over all tasks after learning task $T$, i.e., ACC, which is backpropagated through the tree to update the statistics of the selected nodes. The schedule $S_{best}$ yielding the best ACC score is returned to be used for evaluation on the held-out test sets.

The function GETREPLAYMEMORY($\cdot$) is the policy for retrieving the replay memory $\mathcal{M}$ from the historical data given the task proportion $\boldsymbol{a}$. The number of samples per task determined by the task proportions are rounded up or down accordingly to fill $\mathcal{M}$ with $M$ replay samples in total. The function GETTASKPROPORTION($\cdot$) simply returns the task proportion that is related to given node.

---

**Algorithm 1** Replay Scheduling Monte Carlo tree search

---

**Require:** Tree nodes $v_{1:T}$, Datasets $\mathcal{D}_{1:T}$, Learning rate $\eta$
**Require:** Replay memory size $M$
1:  $\text{ACC}_{best} \leftarrow 0, S_{best} \leftarrow ()$
2:  **while** within computational budget **do**
3:      $S \leftarrow ()$
4:      $v_t, S \leftarrow \text{TREEPOLICY}(v_1, S)$
5:      $v_T, S \leftarrow \text{DEFAULTPOLICY}(v_t, S)$
6:      $\text{ACC} \leftarrow \text{EVALUATEREPLAYSCHEDULE}(\mathcal{D}_{1:T}, S, M)$
7:      $\text{BACKPROPAGATE}(v_t, \text{ACC})$
8:      **if** $\text{ACC} > \text{ACC}_{best}$ **then**
9:          $\text{ACC}_{best} \leftarrow \text{ACC}$
10:         $S_{best} \leftarrow S$
11: **return** $\text{ACC}_{best}, S_{best}$

12: **function** TREEPOLICY$(v_t, S)$
13:     **while** $v_t$ is non-terminal **do**
14:         **if** $v_t$ not fully expanded **then**
15:             **return** EXPANSION$(v_t, S)$
16:         **else**
17:             $v_t \leftarrow \text{BESTCHILD}(v_t)$
18:             $S.\text{append}(\boldsymbol{a}_t)$, where $\boldsymbol{a}_t \leftarrow \text{GETTASKPROPORTION}(v_t)$
19:     **return** $v_t, S$

20: **function** EXPANSION$(v_t, S)$
21:     Sample $v_{t+1}$ uniformly among unvisited children of $v_t$
22:     $S.\text{append}(\boldsymbol{a}_{t+1})$, where $\boldsymbol{a}_{t+1} \leftarrow \text{GETTASKPROPORTION}(v_{t+1})$
23:     Add new child $v_{t+1}$ to node $v_t$
24:     **return** $v_{t+1}, S$

25: **function** BESTCHILD$(v_t)$
26:     $v_{t+1} = \underset{v_{t+1} \in \text{ children of } v}{\arg\max} \max(Q(v_{t+1})) + C\sqrt{\frac{2\log(N(v_t))}{N(v_{t+1})}}$
27:     **return** $v_{t+1}$

28: **function** DEFAULTPOLICY$(v_t, S)$
29:     **while** $v_t$ is non-terminal **do**
30:         Sample $v_{t+1}$ uniformly among children of $v_t$
31:         $S.\text{append}(\boldsymbol{a}_{t+1})$, where $\boldsymbol{a}_{t+1} \leftarrow \text{GETTASKPROPORTION}(v_{t+1})$
32:         Update $v_t \leftarrow v_{t+1}$
33:     **return** $v_t, S$

34: **function** EVALUATEREPLAYSCHEDULE$(\mathcal{D}_{1:T}, S, M)$
35:     Initialize neural network $f_{\boldsymbol{\theta}}$
36:     **for** $t = 1, \dots, T$ **do**
37:         $\boldsymbol{a} \leftarrow S[t-1]$
38:         $\mathcal{M} \leftarrow \text{GETREPLAYMEMORY}(M, \boldsymbol{a})$
39:         **for** $\mathcal{B} \sim \mathcal{D}_t^{(train)}$ **do**
40:             $\boldsymbol{\theta} \leftarrow SGD(\mathcal{B} \cup \mathcal{M}, \boldsymbol{\theta}, \eta)$
41:         $A_{1:T}^{(val)} \leftarrow \text{EVALUATEACCURACY}(f_{\boldsymbol{\theta}}, \mathcal{D}_{1:T}^{(val)})$
42:     $\text{ACC} \leftarrow \frac{1}{T} \sum_{i=1}^{T} A_{T,i}^{(val)}$
43:     **return** ACC

44: **function** BACKPROPAGATE$(v_t, R)$
45:     **while** $v_t$ is not root **do**
46:         $N(v_t) \leftarrow N(v_t) + 1$
47:         $Q(v_t) \leftarrow R$
48:         $v_t \leftarrow$ parent of $v_t$

---

## C  HEURISTIC SCHEDULING BASELINE

NEW
(8z4B)

We implemented a heuristic scheduling baseline to compare against RS-MCTS. The baseline keeps a validation set for the old tasks and replays the tasks which validation accuracy is below a certain threshold. We set the threshold in the following way: Let $A_{t,i}$ be the validation accuracy for task $t$ evaluated at time step $i$. The best evaluated validation accuracy for task $t$ at time $i$ is given by $A_{t,i}^{(best)} = \max(\{A_{t,1}, ..., A_{t,i}\})$. The condition for replaying task $t$ on the next time step is then $A_{t,i} \geq \tau A_{t,i}^{(best)}$, where $\tau \in [0,1]$ is a ratio controlling how much the current accuracy on task $t$ is allowed to decrease w.r.t. the best accuracy. The replay memory is filled with $M/k$, where $k$ is the number of tasks that need to be replayed according to their decrease in validation accuracy. This heuristic scheduling corresponds to the intuition of re-learning when a task has been forgotten. Training on the current task is performed without replay if the accuracy on all old tasks is above their corresponding threshold.

**Grid search for $\tau$.**  We performed a coarse-to-fine grid search for the ratio $\tau$ on each dataset. The best value for $\tau$ is selected according to the highest mean accuracy on the validation set averaged over 5 seeds. The validation set consists of 15% of the training data and is the same for RS-MCTS. We use the same experimental settings as described in Appendix A. The memory sizes are set to $M = 10$ and $M = 100$ for the 5-task datasets and the 10/20-task datasets respectively, and we apply uniform sampling as the memory selection method. We provide the ranges for $\tau$ that was used on each dataset and put the best value in **bold**:

- Split MNIST: $\tau = [0.9, 0.93, 0.95, \mathbf{0.96}, 0.97, 0.98, 0.99]$
- Split FashionMNIST: $\tau = [0.9, 0.93, 0.95, 0.96, \mathbf{0.97}, 0.98, 0.99]$
- Split notMNIST: $\tau = [0.9, 0.93, 0.95, 0.96, 0.97, \mathbf{0.98}, 0.99]$
- Permuted MNIST: $\tau = [0.5, 0.55, 0.6, 0.65, 0.7, \mathbf{0.75}, 0.8, 0.9, 0.95, 0.97, 0.99]$
- Split CIFAR-100: $\tau = [0.3, 0.4, 0.45, \mathbf{0.5}, 0.55, 0.6, 0.65, 0.7, 0.8, 0.9, 0.95, 0.97, 0.99]$
- Split miniImagenet: $\tau = [0.5, 0.6, 0.65, 0.7, \mathbf{0.75}, 0.8, 0.85, 0.9, 0.95, 0.97, 0.99]$

Note that we use these values for $\tau$ on all experiments with Heuristic for the corresponding datasets. The performance for this heuristic highly depends on careful tuning for the ratio $\tau$ when the memory size or memory selection method changes, as can be seen in in Figure 5 and Table 1.

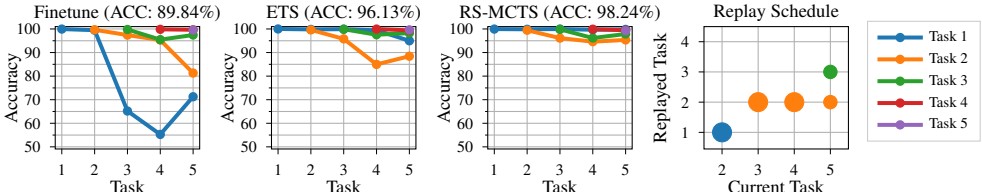

Figure 7: Comparison of test classification accuracies for Task 1-5 on Split MNIST from a network trained without replay (Fine-tuning), ETS, and RS-MCTS. The ACC metric for each method is shown on top of each figure. We also visualize the replay schedule found by RS-MCTS as a bubble plot to the right. The memory size is set to $M = 10$ with uniform memory selection for ETS and RS-MCTS. Results are shown for 1 seed.

## D ADDITIONAL EXPERIMENTAL RESULTS

In this section, we bring more insights to the benefits of replay scheduling in Section D.1 as well as provide metrics for catastrophic forgetting in Section D.2.

### D.1 REPLAY SCHEDULE VISUALIZATION FOR SPLIT MNIST

NEW
(kfup,
dhXe)

In Figure 7, we show the progress in test classification performance for each task when using ETS and RS-MCTS with memory size $M = 10$ on Split MNIST. For comparison, we also show the performance from a network that is fine-tuning on the current task without using replay. Both ETS and RS-MCTS overcome catastrophic forgetting to a large degree compared to the fine-tuning network. Our method RS-MCTS further improves the performance compared to ETS with the same memory, which indicates that learning the time to learn can be more efficient against catastrophic forgetting. Especially, Task 1 and 2 seems to be the most difficult task to remember since it has the lowest final performance using the fine-tuning network. Both ETS and RS-MCTS manage to retain their performance on Task 1 using replay, however, RS-MCTS remembers Task 2 better than ETS by around 5%.

To bring more insights to this behavior, we have visualized the task proportions of the replay examples using a bubble plot showing the corresponding replay schedule from RS-MCTS in Figure 7(right). At Task 3 and 4, we see that the schedule fills the memory with data from Task 2 and discards replaying Task 1. This helps the network to retain knowledge about Task 2 better than ETS at the cost of forgetting Task 3 slightly when learning Task 4. This shows that the learned policy has considered the difficulty level of different tasks. At the next task, the RS-MCTS schedule has decided to rehearse Task 3 and reduces replaying Task 2 when learning Task 5. This behavior is similar to spaced repetition, where increasing the time interval between rehearsals helps memory retention. We emphasize that even on datasets with few tasks, using learned replay schedules can overcome catastrophic forgetting better than standard ETS approaches.

### D.2 ANALYSIS OF CATASTROPHIC FORGETTING

NEW
(dhXe)

In this section, we compare the degree of catastrophic forgetting for our method and the baselines. We use backward transfer (BWT) from Lopez-Paz & Ranzato (2017) for measuring catastrophic forgetting, which is given by

$$\text{BWT} = \frac{1}{T-1} \sum_{i=1}^{T-1} A_{T,i} - A_{i,i}, \tag{3}$$

where $A_{t,i}$ is the test accuracy for task $t$ after learning task $i$. Table 3 shows the ACC and BWT metrics for the experiments in Section 4.3. In general, the BWT metric is consistently better when the corresponding ACC is better. We find an exception in Table 3 on Split CIFAR-100 and Split miniImagenet between Ours and Heuristic with uniform selection method, where Heuristic has better BWT while its mean of ACC is slightly lower than ACC for Ours. Table 4 shows the ACC and BWT metrics for the experiments in Section 4.5, where we see a similar pattern that better ACC yields better BWT. The BWT of RS-MCTS is on par with the other baselines except on Split CIFAR-100 where the ACC on our method was a bit lower than the best baselines.

Table 3: Performance comparison with ACC and BWT metrics for all datasets between RS-MCTS and the baselines with various memory selection methods. The memory size is set to $M = 10$ and $M = 100$ for the 5-task and 10/20-task datasets respectively. We report the mean and standard deviation of ACC and BWT, where all results have been averaged over 5 seeds. RS-MCTS performs better or on par than the baselines on most datasets and selection methods, where MoF yields the best performance in general.

| Selection | Method | Split MNIST | | Split FashionMNIST | | Split notMNIST | |
|---|---|---|---|---|---|---|---|
| | | ACC(%)↑ | BWT(%)↑ | ACC(%)↑ | BWT(%)↑ | ACC(%)↑ | BWT(%)↑ |
| Uniform | Random | 95.91 ± 1.56 | -4.79 ± 1.95 | 95.82 ± 1.45 | -4.35 ± 1.79 | 92.39 ± 1.29 | -4.56 ± 1.29 |
| | ETS | 94.02 ± 4.25 | -7.22 ± 5.33 | 95.81 ± 3.53 | -4.45 ± 4.34 | 91.01 ± 1.39 | -6.16 ± 1.82 |
| | Heuristic | 96.02 ± 2.32 | -4.64 ± 2.90 | 97.09 ± 0.62 | -2.82 ± 0.84 | 91.26 ± 3.99 | -6.06 ± 4.70 |
| | Ours | 97.93 ± 0.56 | -2.27 ± 0.71 | 98.27 ± 0.17 | -1.29 ± 0.20 | 94.64 ± 0.39 | -1.47 ± 0.79 |
| $k$-means | Random | 94.24 ± 3.20 | -6.88 ± 4.00 | 96.30 ± 1.62 | -3.77 ± 2.05 | 91.64 ± 1.39 | -5.64 ± 1.77 |
| | ETS | 92.89 ± 3.53 | -8.66 ± 4.42 | 96.47 ± 0.85 | -3.55 ± 1.07 | 93.80 ± 0.82 | -2.84 ± 0.81 |
| | Heuristic | 96.28 ± 1.68 | -4.32 ± 2.11 | 95.78 ± 1.50 | -4.46 ± 1.87 | 91.75 ± 0.94 | -5.60 ± 2.07 |
| | Ours | 98.20 ± 0.16 | -1.94 ± 0.22 | 98.48 ± 0.26 | -1.04 ± 0.31 | 93.61 ± 0.71 | -3.11 ± 0.55 |
| $k$-center | Random | 96.40 ± 0.68 | -4.21 ± 0.84 | 95.57 ± 3.16 | -7.20 ± 3.93 | 92.61 ± 1.70 | -4.14 ± 2.37 |
| | ETS | 94.84 ± 1.40 | -6.20 ± 1.77 | 97.28 ± 0.50 | -2.58 ± 0.66 | 91.08 ± 2.48 | -6.39 ± 3.46 |
| | Heuristic | 94.55 ± 2.79 | -6.47 ± 3.50 | 94.08 ± 3.72 | -6.59 ± 4.57 | 92.06 ± 1.20 | -4.70 ± 2.09 |
| | Ours | 98.24 ± 0.36 | -1.93 ± 0.44 | 98.06 ± 0.35 | -1.59 ± 0.45 | 94.26 ± 0.37 | -1.97 ± 1.02 |
| MoF | Random | 95.18 ± 3.18 | -5.73 ± 3.95 | 95.76 ± 1.41 | -4.41 ± 1.75 | 91.33 ± 1.75 | -5.94 ± 1.92 |
| | ETS | 97.04 ± 1.23 | -3.46 ± 1.50 | 96.48 ± 1.33 | -3.55 ± 1.73 | 92.64 ± 0.87 | -4.57 ± 1.59 |
| | Heuristic | 96.46 ± 2.41 | -4.09 ± 3.01 | 95.84 ± 0.89 | -4.39 ± 1.15 | 93.24 ± 0.77 | -3.48 ± 1.37 |
| | Ours | 98.37 ± 0.24 | -1.70 ± 0.28 | 97.84 ± 0.32 | -1.81 ± 0.39 | 94.62 ± 0.42 | -1.80 ± 0.56 |

| Selection | Method | Permuted MNIST | | Split CIFAR-100 | | Split miniImagenet | |
|---|---|---|---|---|---|---|---|
| | | ACC(%)↑ | BWT(%)↑ | ACC(%)↑ | BWT(%)↑ | ACC(%)↑ | BWT(%)↑ |
| Uniform | Random | 72.44 ± 1.15 | -25.56 ± 1.39 | 53.99 ± 0.51 | -34.19 ± 0.66 | 48.08 ± 1.36 | -15.98 ± 1.08 |
| | ETS | 71.09 ± 2.31 | -27.39 ± 2.59 | 47.70 ± 2.16 | -41.68 ± 2.37 | 46.97 ± 1.24 | -18.32 ± 1.34 |
| | Heuristic | 76.68 ± 2.13 | -20.82 ± 2.41 | 57.31 ± 1.21 | -30.76 ± 1.45 | 49.66 ± 1.10 | -12.04 ± 0.59 |
| | Ours | 76.34 ± 0.98 | -21.21 ± 1.16 | 56.60 ± 1.13 | -31.39 ± 1.11 | 50.20 ± 0.72 | -13.46 ± 1.22 |
| $k$-means | Random | 74.30 ± 1.43 | -23.50 ± 1.64 | 53.18 ± 1.66 | -35.15 ± 1.61 | 49.47 ± 2.70 | -14.10 ± 2.71 |
| | ETS | 69.40 ± 1.32 | -29.23 ± 1.47 | 47.51 ± 1.14 | -41.77 ± 1.30 | 45.82 ± 0.92 | -19.53 ± 1.10 |
| | Heuristic | 75.57 ± 1.18 | -22.11 ± 1.22 | 54.31 ± 3.94 | -33.80 ± 4.24 | 49.25 ± 1.00 | -12.92 ± 1.22 |
| | Ours | 77.74 ± 0.80 | -19.66 ± 0.95 | 56.95 ± 0.92 | -30.92 ± 0.83 | 50.47 ± 0.85 | -13.31 ± 1.24 |
| $k$-center | Random | 71.41 ± 2.75 | -26.73 ± 3.11 | 48.46 ± 0.31 | -39.89 ± 0.27 | 44.76 ± 0.96 | -18.72 ± 1.17 |
| | ETS | 69.11 ± 1.69 | -29.58 ± 1.81 | 44.13 ± 1.06 | -45.28 ± 1.04 | 41.35 ± 0.96 | -23.71 ± 1.45 |
| | Heuristic | 74.33 ± 2.00 | -23.45 ± 2.27 | 50.32 ± 1.97 | -37.99 ± 2.14 | 44.13 ± 0.95 | -18.26 ± 1.05 |
| | Ours | 76.55 ± 1.16 | -21.06 ± 1.32 | 51.37 ± 1.63 | -37.01 ± 1.62 | 46.76 ± 0.96 | -16.56 ± 0.90 |
| MoF | Random | 77.96 ± 1.84 | -19.44 ± 2.13 | 61.93 ± 1.05 | -25.89 ± 1.07 | 54.50 ± 1.33 | -8.64 ± 1.26 |
| | ETS | 77.62 ± 1.12 | -20.10 ± 1.26 | 60.43 ± 1.17 | -28.22 ± 1.26 | 56.12 ± 1.12 | -8.93 ± 0.83 |
| | Heuristic | 77.27 ± 1.45 | -20.15 ± 1.63 | 55.60 ± 2.70 | -32.57 ± 2.77 | 52.30 ± 0.59 | -9.61 ± 0.67 |
| | Ours | 81.58 ± 0.75 | -15.41 ± 0.86 | 64.22 ± 0.65 | -23.48 ± 1.02 | 57.70 ± 0.51 | -5.31 ± 0.55 |

Table 4: Performance comparison with ACC and BWT metrics for all datasets between RS-MCTS and the baselines in the setting where only 1 sample per class can be replayed. The memory sizes are set to $M = 10$ and $M = 100$ for the 5-task and 10/20-task datasets respectively. We report the mean and standard deviation of ACC and BWT, where all results have been averaged over 5 seeds. RS-MCTS performs on par with the best baselines for both metrics on all datasets except S-CIFAR-100.

| Method | Split MNIST | | Split FashionMNIST | | Split notMNIST | |
|---|---|---|---|---|---|---|
| | ACC(%)↑ | BWT(%)↑ | ACC(%)↑ | BWT(%)↑ | ACC(%)↑ | BWT(%)↑ |
| Random | 92.56 ± 2.90 | -8.97 ± 3.62 | 92.70 ± 3.78 | -8.24 ± 4.75 | 89.53 ± 3.96 | -8.13 ± 5.02 |
| A-GEM | 94.97 ± 1.50 | -6.03 ± 1.87 | 94.81 ± 0.86 | -5.65 ± 1.06 | 92.27 ± 1.16 | -4.17 ± 1.39 |
| ER-Ring | 94.94 ± 1.56 | -6.07 ± 1.92 | 95.83 ± 2.15 | -4.38 ± 2.59 | 91.10 ± 1.89 | -6.27 ± 2.35 |
| Uniform | 95.77 ± 1.12 | -5.02 ± 1.39 | 97.12 ± 1.57 | -2.79 ± 1.98 | 92.14 ± 1.45 | -4.90 ± 1.41 |
| RS-MCTS (Ours) | 96.07 ± 1.60 | -4.59 ± 2.01 | 97.17 ± 0.78 | -2.64 ± 0.99 | 93.41 ± 1.11 | -3.36 ± 1.56 |

| Method | Permuted MNIST | | Split CIFAR-100 | | Split miniImagenet | |
|---|---|---|---|---|---|---|
| | ACC(%)↑ | BWT(%)↑ | ACC(%)↑ | BWT(%)↑ | ACC(%)↑ | BWT(%)↑ |
| Random | 70.02 ± 1.76 | -28.22 ± 1.92 | 48.62 ± 1.02 | -39.95 ± 1.10 | 48.85 ± 1.38 | -14.55 ± 1.86 |
| A-GEM | 64.71 ± 1.78 | -34.41 ± 2.05 | 42.22 ± 2.13 | -46.90 ± 2.21 | 32.06 ± 1.83 | -30.81 ± 1.79 |
| ER-Ring | 69.73 ± 1.13 | -28.87 ± 1.29 | 53.93 ± 1.13 | -34.91 ± 1.18 | 49.82 ± 1.69 | -14.38 ± 1.57 |
| Uniform | 69.85 ± 1.01 | -28.74 ± 1.17 | 52.63 ± 1.62 | -36.43 ± 1.81 | 50.56 ± 1.07 | -13.52 ± 1.34 |
| RS-MCTS (Ours) | 72.52 ± 0.54 | -25.43 ± 0.65 | 51.50 ± 1.19 | -37.01 ± 1.08 | 50.70 ± 0.54 | -12.60 ± 1.13 |

# E    ADDITIONAL FIGURES

**Memory usage.**    We visualize the memory usage for Permuted MNIST and the 20-task datasets Split CIFAR-100 and Split miniImagenet in Figure 8 for our method and the baselines used in the experiment in Section 4.5. Our method uses a fixed memory size of $M = 50$ samples for replay on all three datasets. The memory size capacity for our method is reached after learning task 6 and task 11 on the Permuted MNIST and the 20-task datasets respectively, while the baselines continue incrementing their replay memory size.

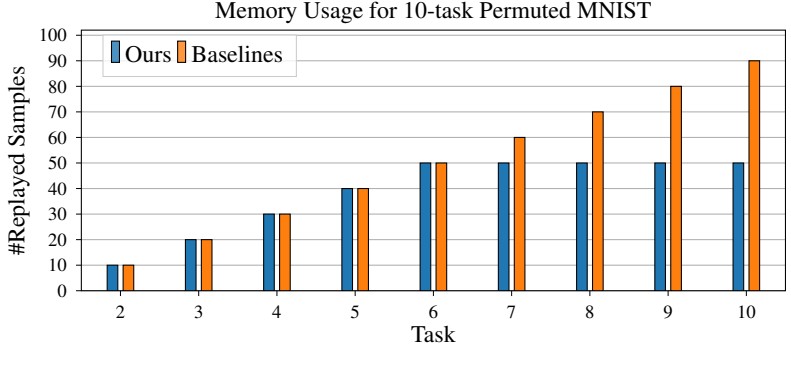

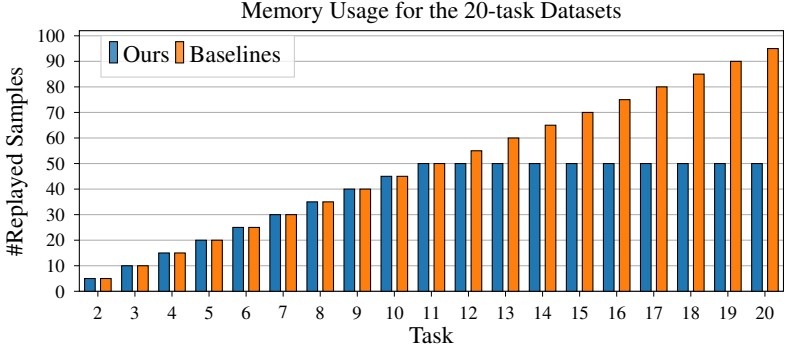

Figure 8: Number of replayed samples per task for 10-task Permuted MNIST (top) and the 20-task datasets in the experiment in Section 4.5. The fixed memory size $M = 50$ for our method is reached after learning task 6 and task 11 on the Permuted MNIST and the 20-task datasets respectively, while the baselines continue incrementing their number of replay samples per task.

