# OpenReview forum: "Learn the Time to Learn: Replay Scheduling for Continual Learning"
_ICLR.cc/2022/Conference — ICLR 2022 Submitted_

### Official Review · Reviewer_8z4B · 2021-10-27

**Correctness:** 4
**Technical Novelty And Significance:** 3
**Empirical Novelty And Significance:** 2
**Recommendation:** 6
**Confidence:** 4

**Main Review:**

I overall found this work interesting and refreshing, since it proposes a novel approach to the improvement of rehearsal-based methods by inquiring what is the best order for revising past tasks. I believe that this issue is very relevant and that there is plenty of room for improvements in this specific area.
I found this paper to be very well written, well organized and easy to understand, for which I congratulate the authors.

However, I find that this work suffers from the following weaknesses:

+ The biggest concern I have with this work is that randomly sampling what information to replay might not be too relevant as a baseline. On the one hand, it is certainly true that this is how the majority of rehearsal approaches work; on the other, it is fairly atypical for these methods to make the assumptions that the authors make here, i.e. that data from past tasks are fully accessible and that the choice of what to replay is up to the model. For this reason, I think it might be appropriate to introduce a somewhat more sophisticated heuristic baseline in addition to the random choice (e.g. keeping a small validation set for past tasks and replaying the ones on which accuracy decreases below a certain level). This would make the experimental section much stronger in my opinion: the finding that the proposed approach outperforms random choice is rather expected in my view.
+ Along the lines of the previous point, I found the initial experiment to be rather unsurprising. Of course, if given the choice when to replay past data, the decision should be made as late as possible: this minimizes forgetting and lets the model come closer to the overall joint trained baseline, similar to [1].
+ I am unsure of the efficiency implications of RS-MCTS in a setting such as continual learning, which is generally quite focused on having a reduced model footprint and execution time. It would make for a very interesting ablation study to see a walltime comparison between it and random replay. I suspect that MCTS might be very demanding, which could naturally bring up the point whether simpler approximations of it could be also proposed for a more effective trade-off.

The following point were minor and did not affect my evaluation
+ I am not sure I correctly understood how robust the proposed policy is to changes in the order of classes: can the findings of Figure 4 generalize to other orders? Can I learn a policy on a specific task order and test it on another?
+ the reference to Figure 5 in the text seems to be missing.
+ While the paper makes it very clear that this approach is effective when the memory buffer is small, what does it happen when we assume that the memory buffer is very large? Do the differences between random and MCTS blur or not? This could be an interesting point to further explore.

[1] GDumb: A Simple Approach that Questions Our Progress in Continual Learning, Prabhu et al.

**Summary Of The Paper:**

The authors propose a reinforcement-learning based approach to mitigate catastrophic forgetting by elaborating on the replay schedule for rehearsal methods. They preliminarily show that not all replay schedules are equally effective and then propose a Monte Carlo Tree Search-based approach that is shown to outperform random replay.

**Summary Of The Review:**

I feel that the paper is relevant and clearly written, however I am insecure about its impact due to the lack of a valid baseline method and its efficiency implications.

---

> ### Author Response · Authors · 2021-11-22
> **Response to Reviewer 8z4B: Additional baselines and contributions**
>
> We thank the reviewer for the valuable feedback and suggestions.
>
> > **"...introduce a somewhat more sophisticated heuristic baseline in addition to the random choice..."**
>
> We are thankful for the suggested baselines for comparison to our RS-MCTS. We have included a baseline using a random replay schedule (named Random) in all experiments in Section 4.1, 4.3, 4.4, and 4.5. We observe that RS-MCTS performs better than Random, especially when the memory size is small.
> Additionally, we implemented a baseline similar to the suggested one that replays the tasks which validation accuracy is below a certain threshold, where this threshold is based on the best achieved validation accuracy on the corresponding task. We present the results for this baseline (named Heuristic) in Section 4.1, 4.3, and 4.4, and describe the method in detail in Appendix C. In Figure 3, we observe that Heuristic often outperforms ETS and serves as a strong baseline to RS-MCTS. However, it highly depends on the threshold selection, as can be seen in Figure 5 where Heuristic since the thresholds selected for Figure 3 does not generalize well when the memory size increases. This further shows the importance of learning the scheduling rather than using pre-defined heuristics. We thank the reviewer again for these baseline suggestions.
>
> > **" unsurprising results...if given the choice when to replay past data, the decision should be made as late as possible [1]..."**
>
> We are glad that the reviewer found the results unsurprising in general as a sign of being convinced of the importance of learning the time to learn.
>
> Moreover, we are dealing with the setting of fixed memory size at each time step. As shown in Section 4.5, we allow the memory size to be much smaller than the total number of tasks. Thus, the setting in the refereed article will fail and this shows the importance of scheduling in the general continual learning (CL) setting.
>
> > **"...would make for a very interesting ablation study to see a walltime comparison between it and random replay... MCTS might be very demanding"**
>
> We agree that MCTS is computationally demanding time-wise. However, our intention of this work was to illustrate the importance of learning when to replay which tasks. As discussed in the future work (last paragraph), we will explore learning these policies using reinforcement learning (RL) which can make replay scheduling applicable to different CL tasks without having to re-learn the policy. But since generalization in RL is an active research field by itself, we wanted to focus on the contributions with showing the learning replay schedules is important in realistic settings by evaluating the idea on experiments with CL benchmarks.
>
> > **"Reference to Figure 5 in the text seems to be missing."**
>
> We have corrected this in the revised version.
>
> > **"...what happens when we assume that the memory buffer is very large? Do the differences between random and MCTS blur or not?"**
>
> As shown in Figure 5, if the memory is very large, the difference indeed blur. However, in large systems in real-world, the number of tasks may be huge and the memory can potentially never be large enough.

---

> > ### Comment · Reviewer_8z4B · 2021-11-24
> > **Thank you for your response**
> >
> > I am thankful to the authors for taking the time to consider my suggestion and integrating them in their work. I am most pleased of their inclusion of a Heuristic baseline, which I find makes the experimental section much stronger and more convincing, further shedding light on the value added by tackling replay schedule in a learned way.
> >
> > For this reason I am increasing my original evaluation and suggest acceptance.

---

### Official Review · Reviewer_gm3R · 2021-11-01

**Correctness:** 3
**Technical Novelty And Significance:** 3
**Empirical Novelty And Significance:** 2
**Recommendation:** 5
**Confidence:** 4

**Main Review:**

Learning to schedule replay for continual learning is an interesting and novel idea. Clear writing helps readers appreciate the importance of this topic and understand the proposed solution. The behaviour of the method is well illustrated with the analysis in Fig. 4.  Unfortunately, the experiments fall short of showing the benefit of the solution in a more realistic setting.

Here, the paper focuses more on memory efficiency and ignore computing efficiency. It seems that training with MCTS is more expensive as multiple trials need to be done (around 40 iterations to converge-Fig. 3) in exchange for a few % of final accuracy improvement. Moreover, the performance gap reduces if more sophisticated sampling techniques are employed (Table 1). I wonder if there is still improvement as MCTS is applied to recent replay methods [1,2,3]. Also, as M  increases (e.g. up to 100-Fig. 5), the performance gap is marginal. Here, we need an explanation of the practicality of the trade-off between training and memory complexity (i.e. 1 iteration+M=100 vs 40 iterations+M=10, which one is better?).

Another concern is the need for a complicated model like MCTS if we can perform multiple runs. In Fig. 3, some tasks only need one iteration (i.e. the found policy is random) to achieve significantly better results. To prove the benefit of MCTS, the authors need to compare MCTS with simple baselines such as random search and/or standard UCB bandit algorithms.



[1] Riemer, Matthew, Ignacio Cases, Robert Ajemian, Miao Liu, Irina Rish, Yuhai Tu, and Gerald Tesauro. "Learning to Learn without Forgetting by Maximizing Transfer and Minimizing Interference." In International Conference on Learning Representations. 2018.
[2] Buzzega, Pietro, Matteo Boschini, Angelo Porrello, Davide Abati, and Simone Calderara. "Dark Experience for General Continual Learning: a Strong, Simple Baseline." In NeurIPS. 2020.
[3] Chaudhry, Arslan, Albert Gordo, Puneet Dokania, Philip Torr, and David Lopez-Paz. "Using Hindsight to Anchor Past Knowledge in Continual Learning." In Proceedings of the AAAI Conference on Artificial Intelligence, vol. 35, no. 8, pp. 6993-7001. 2021.

**Summary Of The Paper:**

This paper proposes a new continual learning method that learns to select samples for the replaying process. The replay memory is filled with samples from previous tasks according to a specific proportion corresponding to the action performed at the current task. The paper proposes to find the optimal sequence of actions by using Monte-Carlo Tree Search (MCTS) with the reward as the average accuracy over all tasks after seeing the final task. The experiments demonstrate that MCTS improves the continual learning performance especially when the memory size is small and is compatible with some sample-selection strategies.

**Summary Of The Review:**

Overall, I like the topic and proposed method. However, the exepriments need improvement. I may consider raising my score if the authors can: (1) clarify the practicality of the trade-off mentioned above, (2) combine MCTS with recent replay techniques and show clear improvement, and (3)  prove that MCTS is better than simple search/optimization methods.

---

> ### Author Response · Authors · 2021-11-22
> **Response to Reviewer gm3R: Contributions and benefit of MCTS**
>
> We thank the reviewer for the valuable feedback and for finding our idea interesting.
>
> > **"...realistic setting ...  ignore computing efficiency."**
>
> We agree about the concern regarding MCTS as multiple rollouts are needed for each task and discussed this in the future work that learning a policy using e.g. reinforcement learning (RL) that generalize among different continual learning (CL) tasks will sidestep this problem. However, generalization in RL is an active research field by itself [1] which may make it harder to disentangle the contributions of this paper.  Thus, in this work, we mainly want to show the two contributions regarding the new CL setting and the importance to learn the scheduling of what task to replay at different times.
>
> [1] Kirk, R., Zhang, A., Grefenstette, E., & Rocktaschel, T. (2021). A Survey of Generalisation in Deep Reinforcement Learning. https://arxiv.org/abs/2111.09794
>
> > **"To prove the benefit of MCTS, the authors need to compare MCTS with simple baselines such as random search and/or standard UCB bandit algorithms."**
>
> We believe that there is a potential misunderstanding. We mainly want to prove the benefit of scheduling which task to replay rather than proving the benefit of MCTS. As shown in the experiments, we used breadth-first search (BFS) which provides optimal scheduling traversing all possible scheduling when the search space is small. Since BFS does not scale well with more than 5 tasks, MCTS is a method that allows learning schedules for longer task horizons which allows us to show the importance of scheduling in larger CL tasks. Other search algorithms which scales are indeed possible alternatives to MCTS and will be able to show our contribution in similar ways.

---

> > ### Comment · Reviewer_gm3R · 2021-11-24
> > **Reply to Authors**
> >
> > Thank you for your response.
> >
> > While I understand the goal of the paper is more about introducing a new setting, in its current form, I don't think the contribution is significant enough for publication.
> >
> > As said before, without a detailed analysis of the trade-off, we cannot see the practical advantage of learning a replay schedule. Also, since MCTS is your proposed method to solve the problem, it is necessary to compare it with other alternatives. One may question why we should implement the complicated MCTS if random replaying is good enough? A fair comparison will standard baselines will strengthen the paper's experiments and make the contribution clearer.

---

> > > ### Author Response · Authors · 2021-11-27
> > > **Response to Reviewer gm3R: Additional results with recent replay methods and goal of paper**
> > >
> > > Thanks for the discussion.
> > >
> > > > **...compare it with other alternatives... random replaying is good enough...**
> > >
> > > Thanks for the feedback. In addition to ETS and its combination with k-means, k-center, MoF, as well as heuristic scheduling, A-GEM, ER-ring, we have added more experiments where we compare to more recent replay methods as was suggested.
> > > The new experiments include comparing our proposed replay scheduling in combination with Meta-Experience Replay (MER), Dark Experience Replay (DER), and DER++ as suggested. In general, we combine MER, DER, and DER++ with RS-MCTS and compare to the methods without scheduling.
> > >
> > > We attach the experimental results with Split MNIST below. The memory size is set to M=10 and the results are averaged over 5 seeds. We will add results for all datasets in the revised version of our paper.
> > >
> > > |        |                |       Split MNIST       |                          |
> > > |--------|----------------|:-----------------------:|:------------------------:|
> > > | **Method** | **Schedule**       |           **ACC**           |            **BWT**           |
> > > | MER    | Random         |      93.07 +/- 0.81     |      -8.36 +/- 0.99      |
> > > |   -"-     | ETS            |      92.97 +/- 1.73     |      -8.52 +/- 2.15      |
> > > |   -"-     | RS-MCTS (Ours) | **96.44 +/- 0.72** |  **-4.14 +/- 0.94** |
> > > | DER    | Random         |      98.23 +/- 0.53     |      -1.89 +/- 0.65      |
> > > |    -"-    | ETS            |     98.17 +/- 0.35,     |      -2.00 +/- 0.42      |
> > > |    -"-    | RS-MCTS (Ours) | **99.02 +/- 0.10** | **-0.91 +/- 0.13** |
> > > | DER++  | Random         |      97.90 +/- 0.52     |      -2.32 +/- 0.67      |
> > > |    -"-     | ETS            |      97.98 +/- 0.52     |      -2.24 +/- 0.66      |
> > > |   -"-      | RS-MCTS (Ours) | **98.84 +/- 0.21** |  **-1.14 +/- 0.26** |
> > >
> > > Using MCTS to schedule the memory significantly outperforms the method with ETS or random scheduling.
> > > The results again confirm that learning the time to learn is important for the continual learning performance given the same memory constraints and it can benefit any existing continual learning framework.
> > >
> > > > **...the goal of the paper is more about introducing a new setting...**
> > >
> > > This is indeed one of our goals where the new setting adjusts the current continual learning research closer to real-world needs.
> > > However, we would like to enhance our second contribution which is to show the importance of learning the time to learn, aka, how to schedule the memory is critical for the continual learning performance. In this paper, we use MCTS as an example to demonstrate such importance and show significant improvements compared to methods without learning the replay schedule given the same memory budget.
> > >
> > > Please let us know if you have any further questions.

---

> > > > ### Author Response · Authors · 2021-11-29
> > > > **Response to Reviewer gm3R: Additional results part 2**
> > > >
> > > > Dear Reviewer gm3R,
> > > >
> > > > In addition to the results above, we have further added the results with  Hindsight  Anchor  Learning  (HAL), as well as Permuted MNIST dataset. The current updated results are shown in the table below. The advantage of our method remains with the combination of different CL methods across different datasets.
> > > >
> > > > |        |                |       Split MNIST       |                          |
> > > > |--------|----------------|:-----------------------:|:------------------------:|
> > > > | **Method** | **Schedule**       |           **ACC**           |            **BWT**           |
> > > > | HAL    | Random         |      97.24 +/- 0.70     |      -2.77 +/- 0.90      |
> > > > |   -"-     | ETS            |      97.21 +/- 1.25     |      -2.80 +/- 1.59      |
> > > > |   -"-     | RS-MCTS (Ours)           |      97.96 +/- 0.15     |      -1.85 +/- 0.18      |
> > > > | MER    | Random         |      93.07 +/- 0.81     |      -8.36 +/- 0.99      |
> > > > |   -"-     | ETS            |      92.97 +/- 1.73     |      -8.52 +/- 2.15      |
> > > > |   -"-     | RS-MCTS (Ours) | **96.44 +/- 0.72** |  **-4.14 +/- 0.94** |
> > > > | DER    | Random         |      98.23 +/- 0.53     |      -1.89 +/- 0.65      |
> > > > |   -"-     | ETS            |     98.17 +/- 0.35     |      -2.00 +/- 0.42      |
> > > > |   -"-     | RS-MCTS (Ours) | **99.02 +/- 0.10** | **-0.91 +/- 0.13** |
> > > > | DER++  | Random         |      97.90 +/- 0.52     |      -2.32 +/- 0.67      |
> > > > |   -"-     | ETS            |      97.98 +/- 0.52     |      -2.24 +/- 0.66      |
> > > > |   -"-     | RS-MCTS (Ours) | **98.84 +/- 0.21** |  **-1.14 +/- 0.26** |
> > > >
> > > > |        |                |      Permuted MNIST     |                          |
> > > > |--------|----------------|:-----------------------:|:------------------------:|
> > > > | **Method** | **Schedule**      |           **ACC**           |            **BWT**           |
> > > > | HAL    | Random         |      88.49 +/- 0.99     |      -7.03 +/- 1.05      |
> > > > |   -"-     | ETS            |      88.46 +/- 0.86     |      -7.26 +/- 0.90      |
> > > > |   -"-     | MCTS           |      89.14 +/- 0.74     |      -6.29 +/- 0.74      |
> > > > | MER    | Random         |      75.90 +/- 1.34     |      -21.69 +/- 1.47     |
> > > > |   -"-     | ETS            |      73.01 +/- 0.96     |      -25.19 +/- 1.10     |
> > > > |    -"-    | RS-MCTS (Ours) | **79.72 +/- 0.71** | **-17.42 +/- 0.78** |
> > > > | DER    | Random         |      87.51 +/- 1.10     |      -8.81 +/- 1.28      |
> > > > |   -"-     | ETS            |      85.71 +/- 0.75     |      -11.15 +/- 0.87     |
> > > > |    -"-    | RS-MCTS (Ours) | **90.11 +/- 0.18** |  **-5.89 +/- 0.23** |
> > > > | DER++  | Random         |      87.89 +/- 1.10     |      -8.35 +/- 1.33      |
> > > > |   -"-     | ETS            |      85.25 +/- 0.88     |      -11.60 +/- 1.03     |
> > > > |   -"-     | RS-MCTS (Ours) | **89.84 +/- 0.22** |  **-6.13 +/- 0.29** |
> > > >
> > > > We will add results for all datasets in the revised version of our paper.

---

### Official Review · Reviewer_dhXe · 2021-11-03

**Correctness:** 3
**Technical Novelty And Significance:** 2
**Empirical Novelty And Significance:** 2
**Recommendation:** 3
**Confidence:** 4

**Main Review:**

1. As the authors already mentioned, there are many relevant methods that adaptively select samples (rather than task-uniform selection) to memorize in an on-/offline manner during continual learning. A naive comparison with ETS is not attractive to validate the effectiveness of the proposed method and it requires in-depth comparisons with recent adaptive sample selection-based works.

2. The buffer should memorize a larger number of past tasks' instances to control the instance occupation per task. It enables the model to reuse the discarded instances at previous time steps, which is not allowed standard replay-based continual learning scenario.

3. The requirement of the validation dataset is also the weakness of the proposed method. The authors store and use 15% of training instances as a validation during the continual learning process. This is essential for computing scheduling rewards. To this end, the proposed method indeed needs more room to memorize previous samples compared to baselines. To this end, I'm quite negative about fairness in comparison.

4. In general, there is a lack of analyses in the proposed method, such as
- the size of the validation set,
- the possible design of rewards,
- analysis of catastrophic forgetting,
- continual learning settings including class-incremental learning,
- the change of per-task performance according to the selection of adaptive task-selection schedule,
- the classwise performance when the size of M is smaller than the number of classes like the case of Section 4.5 (i.e., the case that the buffer cannot cover all classes as a replay),
...

**Summary Of The Paper:**

The paper proposes a new scheduling technique for constructing the replay buffer during training. As different to existing baselines which use a task-equal selection, they suggest a dynamic selection of the past tasks' replay exemplars, based on the observation that the time to revisit past tasks affects the averaged performance of the continual learner. The technique is simple yet seems to be effective compared to task-equivalent selection schedule.


**Summary Of The Review:**

Motivation is reasonable and the proposed technique is simple yet seems to be effective compared to a task-equivalent selection schedule. However, there are several concerns.

---

> ### Author Response · Authors · 2021-11-22
> **Part 1/2, Response to Reviewer dhXe: Contribution, fairness when using validation dataset**
>
> We thank the reviewer for the valuable feedback.
>
> > **(1) "...recent adaptive sample selection-based works."**
>
> We believe that might have been a misunderstanding of the of related work. There is no prior work on *adaptive* sample selection method in terms of scheduling. To the best of our knowledge, we are the first work that proposes learning the time to learn. The related works on *sample selection* methods that we have mentioned in Section 2 are about selecting high quality samples for populating the memory given the scheduling method such as ETS.
>
> Moreover, as discussed in the paper, our method can be combined with different existing memory selection methods to improve the sample quality given a schedule. We have shown these results in Table 1 where our method is combined with k-means, k-center, and Mean-of-Features (MoF). Finally, we have also compared to the state-of-art A-GEM and ER-Ring where only 1 sample per class is available for replay in Section 4.5.
>
> > **(2) "...It enables the model to reuse the discarded instances at previous time steps, which is not allowed standard replay-based continual learning scenario."**
>
> Yes. This is indeed different from the standard replay-based continual learning (CL) setting. We would like to argue that this is one of our contribution to propose a new CL setting that is better aligned with the real-world need, since real-world companies stores historical data rather than removing them. However, general CL is still needed as data indeed comes continuously and the amount of compute is limited.
>
>
> > **(3) "The requirement of the validation dataset is also the weakness of the proposed method...needs more room to  memorize  previous  samples.. fairness in comparison"**
>
> This might be a misunderstanding. In our proposed CL setting, the historical data are available and the memory is limited due to limitations on compute. Thus, validation sets are not additional memories.
>
> To address your concern about the fairness in the comparisons between RS-MCTS and the baselines. We re-ran the experiments with ETS where training is done on both train+val sets, i.e., the original training sets for each dataset, such that they use the same amount of data as RS-MCTS. We have updated the Figure 3 and 5 as well as Table 1 with the new ETS baseline Results in the revised version. Note that the conclusions remain the same that replay scheduling outperforms the ETS baseline, especially for small memory sizes.

---

> > ### Author Response · Authors · 2021-11-22
> > **Part 2/2, Response to Reviewer dhXe: Suggested experimental analyzes**
> >
> > > **(4) "...analyses in the proposed method, such as: Size of the validation set, Possible design of rewards, Analysis of catastrophic forgetting, CL settings including class-incremental learning etc"**
> >
> > **Size of the validation set and  Possible design of rewards.** Thank you for the suggestion. Instead of using validation sets for computing rewards, we can also use arbitrary size of training data to compute rewards. The design of the rewards are mainly driven by CL goal in the paper, thus, we used the averaged final performance over all tasks (ACC). An alternative reward can be the average accuracy over both tasks and time. Our method can handle different reward designs providing scheduling for different CL goals.  However, we would like to note that these settings are orthogonal to our contribution revisited in the general reply.
> >
> > **Analysis of catastrophic forgetting.** We have added Table 3 and 4 in Appendix D.1 which shows the backward transfer (BWT) metric from [1]. In Table 3, our method receives higher BWT scores than the baselines with a few exceptions on Split CIFAR-100 and Split miniImagenet when using uniform and k-means for memory selection. In Table 4, the BWT of our method is on par with the other baselines except on Split CIFAR-100, where the ACC on our method was a bit lower. The conclusions of the work remains the same that learning replay schedules is important in our proposed CL setting.
> >
> >
> >
> > **CL settings including class-incremental learning.** Our problem setting can actually be used for class-incremental learning (Class-IL) with some slight changes in the replay scheduling tree described in Section 3.2 and Figure 2. For example, if we always make sure to replay a few examples from all tasks, we help the neural networks to not forget the old tasks completely which happens in Class-IL settings for Finetuning baselines. Then, we can let our method select from which tasks we should fill the remaining parts in the memory at different times. However, the Class-IL setting may lead to very large memory sizes in later stages in the real-world setting. Thus, we focus on the fixed memory size setting that aligns better with real-world needs which is one of the contributions of the work.
> >
> > **Change of per-task performance according to the selection of adaptive task-selection schedule.** We have added Figure 7 in Appendix D.1 that illustrates the benefits of learning replay schedules compared to using ETS by visualizing the per-task performances. The results show that our method manages to retain the the task performances better than ETS by replaying difficult tasks more than others.
> >
> > **Classwise performance in Section 4.5.** This would have been interesting to look deeper into, but we did not have time unfortunately.
> >
> > We thank the reviewer for the suggested analyses to improve the experiment section.
> >
> > References:
> > [1] Lopez-Paz, D., & Ranzato, M. A. (2017). Gradient episodic memory for continual learning. Advances in neural information processing systems, 30, 6467-6476.

---

> > > ### Comment · Reviewer_dhXe · 2021-11-27
> > > **Reply to authors**
> > >
> > > Thank you for your response and I admire the additional results and more ablation study during the rebuttal period.
> > >
> > > I thoroughly read the responses and reviews from other reviewers, but I still do not think the submission is sufficient to be published in this venue. I believe that the suggested new problem setup where the buffer should **memorize all past tasks' datasets** (**a learner can access all of the previous data at any time**) is too restricted, which is only applicable to a special case of continual learning scenario and is opposite to a general definition of continual learning. And the setting seems to be more similar to a curriculum learning setting rather than continual learning.  Overall, the goal of this submission is reasonable but the problem that the submission would tackle is neither attractive to me nor generally applicable.
> > >
> > > I'll keep my original score.

---

> > > > ### Author Response · Authors · 2021-11-29
> > > > **Response to Reviewer dhXe: Real-world applicability of our proposed new CL setting**
> > > >
> > > > Thanks for the discussion.
> > > >
> > > > > **"...setting more similar to a curriculum learning setting rather than continual learning..."**
> > > >
> > > > Thanks for the question. We would like to clarify some differences and similarities of our proposed setting with curriculum learning and continual learning.
> > > >
> > > > Curriculum learning tries to learn the task order with the setting that the training data is **static** and it learns a schedule to train the model to obtain better performance for a fixed set of tasks.
> > > > This is completely different from either traditional continual learning or our proposed new continual learning setting where the fundamental assumption is that the data comes in streams with new tasks associated. We do not control how data and tasks come.
> > > >
> > > > Comparing our setting with the traditional continual learning setting, we share the fundamental setting that the data comes in streams and we cannot re-train on all historical data. Also, we share the common goal that the model should do well both in tasks with historical data and tasks associated with new data. In memory-based continual learning, we also share the same constraints that the memory size is limited where we argue that this limitation is mainly associated with compute instead of storage which aligns with the real-world where data storage is cheap and easy but retraining large ML models is computationally expensive where the time would not allow. The only difference is that we allow filling this limited memory from historical data or another external memory. Here, we argue that historical data are never thrown away in real-world settings. Thus, to make continual learning align with real-world needs, we should keep the limited memory assumption for training but allow access to historical data to fill this memory.
> > > >
> > > > Moreover, we would also like to enhance our second contribution which is learning the time to learn. This shows the importance of replay scheduling in continual learning which aligns with human learning.
> > > >
> > > > > **"...memorize all past tasks' datasets"**
> > > >
> > > > As discussed above, we assume that the agent has access to historical data to fill the memory instead of "memorize all past" datasets in the sense of using all historical data in the memory during the training process. The memory that can be used for training is very small due to computational limitations.
> > > >
> > > > > **"... suggested new problem setup is too restricted... only applicable to a special case of continual learning scenario..."** and **"...the problem that the submission would tackle is not generally applicable."**
> > > >
> > > > Instead of "restricted", on the contrary, the suggested new setup as one of our contributions enables continual learning research to make an impact in all real-world scenarios which this field is trying to solve. In the real-world, almost all companies store their historical data. Please let us know from a real-world impact perspective if you have any questions regarding the general applicability of our setting.

---

### Official Review · Reviewer_kfup · 2021-11-04

**Correctness:** 3
**Technical Novelty And Significance:** 3
**Empirical Novelty And Significance:** Not applicable
**Recommendation:** 8
**Confidence:** 4

**Main Review:**

### **Questions**
- It is not clear if the memory selection process through approaches like k-means chooses datapoints or is it a replacement to MCTS for choosing a sequence of tasks? I have assumed the former while reading the paper.
- How does this algorithm scale in life-long learning paradigms. Are the K (number of branches) models stored in memory, so it is easier to increase the number of models from T to T+1 without training from scratch? Or am I getting something wrong?

### **Strengths**
- All the contributions ion the introduction have been verified empirically on 6 reasonably diverse datasets.
- Table 1 is quite interesting, as it shows a significant boost due to an optimal selection of tasks for replay memory strongly supporting the claim that replay schedule can be quite important for continual learning
- For extreme scenarios where the replay memory does not have some of the classes that were previously seen, the approach of selecting the correct schedule is able to match performance for continual learning baselines that were designed specifically for memory-efficient training.
- The paper (along with the appendix) is generally well written and mostly self-contained.

### **Weaknesses** and **Suggestions**
- The concerns I have with using MCTS to search for an optimal replay sequence are as follows:
  - The time taken to run the training for each replay sequence from the root to leaf node is equal to the time taken to train all the T tasks. And this process has to be done K number of times to get enough branches for a decent estimate. O(TK) training time does not sound very appealing when K or T is very high.
  - As T increases, the value of K might need an update, as the tree will progressively become sparser. At a high enough T, wouldn't the greediness of the approach result in the suboptimality of the replay sequence? To prevent that K would need to be increased, which is often possible using a lot of parallel compute, but that is also limited to the processing constraints posed on the problem in the introduction.
- In the bubble plot in Figure 4, the explanation is that tasks 4 and 6 need lesser replay in the early stages probably because they are less correlated with the other tasks. I would argue that if two tasks are correlated, then there is a greater need to replay that information to expose the model to high-quality negative samples (negative samples that are close to the positive/anchor points; hence the decision boundary is more certain). Either way, It would have been interesting to see a bubble plot for a smaller and analyzable dataset such as MNIST. Examples like the tasks for classifying digits 8, 0 and 3 which can often be confused would have been an interesting analysis tool.
- There is potential inconsistencies between Figure 5 plots and Figure 3. For M=100 in Figure 5, the accuracies for both ETS and Ours on SplitImagenet and SplitCIFAR are quite different from the ones reported in Figure 3.


**Summary Of The Paper:**

The key motivation of this work is that the bottleneck of replay in continual learning is the processing time in each training cycle and not storage space for the historical dataset. Hence, this work has been approached from the angle of fixed-sized memory allowance for each experience training cycle. The main research question of this work is: Can the replay schedule (i.e. which task is replayed is which time) significantly affect the training process over time? After demonstrating the effect of the replay schedule, a monte carlo tree search is introduced as a methodology to learn an optimal replay schedule for a series of tasks. This approach shows improvement over naive selection process of replay memory as well as gives the flexibility to apply any selection process for individual sample points for each task. Finally, the efficiency of this approach is tested in extreme scenarios where the replay memory is smaller than the number of classes (i.e. in the training cycles, samples of some classes will not be sampled). Experiments on 6 datasets (varying over 5, 10, and 20 tasks) is used as empirical proof of performance.

**Summary Of The Review:**

While the ideas in this paper are sound and are justified with ample experiments, I have some concerns about how this approach would scale with a significantly larger number of tasks, which was one of the main motivations of the work. Hence my initial rating is borderline

---

> ### Author Response · Authors · 2021-11-22
> **Response to Reviewer kfup: Contribution, corrected inconsistencies, and additional insights**
>
> We thank the reviewer for the valuable feedback and suggestions.
>
> > **“...concerns about how this approach would scale with a significantly larger number of tasks, which was one of the main motivations of the work.”** and **“...concerns I have with using MCTS to search for an optimal replay sequence…”**
>
> We believe that there is a potential misunderstanding regarding the main contribution of our paper. Rather than proposing an approach applicable to life-long learning settings, our main goal was to demonstrate the importance of learning which tasks to replay at different time steps in a continual learning (CL) setting. As a way of illustrating this idea, we use Monte Carlo Tree Search (MCTS) as an example method for learning a policy that suggests which tasks to replay and its sample proportion in the replay memory at different time steps. To the best of our knowledge, this work is an initial step for developing replay approaches that consider the time to replay in CL, and we hope to encourage more research in this direction.
>
> We agree on the raised concerns regarding scalability to larger number of tasks, training time and the number of MCTS iterations required. As we have discussed in the future work (last paragraph), methods such as reinforcement learning will be able to learn more general policies that can scale to larger action spaces and generalize among different CL learning tasks. As generalization of RL is a research field by itself, we would like to disentangle our contribution from policy learning for this work and focus on the new CL setting we proposed and showing the importance of memory scheduling.
>
> > **“There is potential inconsistencies between Figure 5 plots and Figure 3...”**
>
> We thank the reviewer for pointing out these inconsistencies. The reason for this was due to differences in the implementations of how the replay memory is sampled in the experiments in Figure 3 (Section 4.1) and Figure 5 (Section 4.3). We re-ran the experiments for Figure 5 and updated this figure with the new results to make the replay memory sampling consistent throughout the paper. We thank the reviewer again for notifying us on these differences between Figure 3 and Figure 5.
>
> > **“In the bubble plot in Figure 4, … I would argue that if two tasks are correlated, then there is a greater need to replay that information to expose the model to high-quality negative samples”**
>
> We thank the reviewer for the additional insights. We agree that replaying correlated tasks more will probably create more clear decision boundaries between the classes in these tasks. On the other hand, if two tasks are correlated, replaying both is potentially not needed. The reason for showing Figure 4 is to demonstrate that the optimal replay schedule is highly non-linear and shows similar behavior as scheduling for spaced repetition learning techniques which is consistent with real-world children education.
>
> > **“... It would have been interesting to see a bubble plot for a smaller and analyzable dataset such as MNIST...”**
>
> We thank the reviewer for this suggestion and have added a similar bubble plot visualization for Split MNIST with analysis in Figure 7 in Appendix D.1. Furthermore, we demonstrate how learning the replay schedule helps retaining more difficult tasks by replaying them more than others.

---

### Author Response · Authors · 2021-11-22
**Overall Response from Authors to Reviewers: Recap of Contributions and Info on Revised Paper**

We want to thank all reviewers for their constructive feedback and suggestions. We appreciate that Reviewer **kfup**, **gm3R**, and **8z4B** found the idea with replay scheduling interesting and said that the paper is well-written.

We believe that some concerns might be related to misunderstandings regarding our contributions, thus we revisit our contribution below.

**Contributions:** We challenge the current continual learning (CL) setting and propose a new CL setting where historical data is available while the amount of compute is limited at any time. This new setting aligns better with real-world needs given that new data come continuously but companies never throw away historical data.  Moreover, we propose learning the time to learn, i.e., learning which tasks to replay at different times. We show the importance of memory scheduling, which has been overlooked by the current literature, by using MCTS as an examplar method through evaluation on several benchmark datasets in the CL literature.

We have revised the manuscript and color-coded the changes in **blue**. Furthermore, we have added references in the margin to the reviewer who requested the fix or add-on where it was appropriate.

We have replied to each reviewer separately below in the comment section. Please let us know if you have any further questions.

---

### Author Response · Authors · 2021-11-24
**Thanks**

We thank reviewer 8z4B for the valuable discussion and increase of the score.
Reviewer dhXe and gm3R, please let us know if you have any other questions, and we are happy to provide further clarification.

---

### Decision · Program_Chairs · 2022-01-20

**Decision:**

Reject

**Comment:**

This paper studies the problem of dynamically selecting samples to replay given that all previous data is stored. The paper shows that in this setting, selecting which samples to replay outperforms several baselines over a variety of datasets.

I believe that the reviewers understood this work, but their initial opinions were quite mixed.

Two of the reviewers did not "accept" this setting (all past data stored and accessible) as a reasonable one for continual learning. The discussion did not lead to a reconciliation.

I found truth in both views. On one side, I can believe that the proposed setting has applications (recommender systems where historical data is kept seems like a reasonable one). I also find the approach reasonable since "compute" is often the bottleneck and not memory/storage. On the other, I also see that this is specializing the CL problem a bit and so, while immediately useful, may or may not help to improve more general continual-learning approaches. This is highly speculative. Another argument against this setting is that it is not absolutely clear that in this setting CL approaches are necessarily required. This really depends on the specifics of the problems.

Several of the questions and weaknesses discussed by other reviewers were also discussed and addressed by the reviewers.

Overall, the final score from the reviewers makes this a very borderline paper. Further, even amongst the positive reviewers, one provides an overall recommendation of a 6 (marginally above the acceptance threshold). In the end, the paper was in the category of papers that were examined closely for possible acceptance, but the broad view of the area chair and the reviewers was that the paper could benefit from additional work before publication.